# DISTILLED DECODING 1: ONE-STEP SAMPLING OF IMAGE AUTO-REGRESSIVE MODELS WITH FLOW MATCHING

**Enshu Liu**,[*] **Xuefei Ning, Yu Wang,**
Department of EE, Tsinghua University
les23@mails.tsinghua.edu.cn
foxdoraame@gmail.com
yu-wang@mail.tsinghua.edu.cn

**Zinan Lin**[†]
Microsoft Research
zinanlin@microsoft.com

## ABSTRACT

Autoregressive (AR) models have recently achieved state-of-the-art performance in text and image generation. However, their primary limitation is slow generation speed due to the token-by-token process. We ask an ambitious question: can a pre-trained AR model be adapted to generate outputs in just *one or two steps*? If successful, this would significantly advance the development and deployment of AR models. We notice that existing works that attempt to speed up AR generation by generating multiple tokens at once fundamentally cannot capture the output distribution due to the conditional dependencies between tokens, limiting their effectiveness for few-step generation. To overcome this, we propose `Distilled Decoding` (`DD`), which leverages flow matching to create a deterministic mapping from Gaussian distribution to the output distribution of the pre-trained AR model. We then train a network to distill this mapping, enabling few-step generation. The entire training process of `DD` does *not* need the training data of the original AR model (as opposed to some other methods), thus making `DD` more practical. We evaluate `DD` on state-of-the-art *image* AR models and present promising results. For VAR, which requires 10-step generation (680 tokens), `DD` enables one-step generation ($6.3\times$ speed-up), with an acceptable increase in FID from 4.19 to 9.96 on ImageNet-256. Similarly, for LlamaGen, `DD` reduces generation from 256 steps to 1, achieving an $217.8\times$ speed-up with a comparable FID increase from 4.11 to 11.35 on ImageNet-256. In both cases, baseline methods completely fail with FID scores $>$100. `DD` also excels on *text-to-image generation*, reducing the generation from 256 steps to 2 for LlamaGen with minimal FID increase from 25.70 to 28.95. As the first work to demonstrate the possibility of one-step generation for image AR models, `DD` challenges the prevailing notion that AR models are inherently slow, and opens up new opportunities for efficient AR generation. The code and the pre-trained models will be released at https://github.com/imagination-research/distilled-decoding. The project website is at https://imagination-research.github.io/distilled-decoding.

## 1 INTRODUCTION

Autoregressive (AR) models (Van den Oord et al., 2016; Chen et al., 2018; Esser et al., 2021; Razavi et al., 2019; Lee et al., 2022; Yu et al., 2021; Chang et al., 2022; Li et al., 2023; 2024a; Touvron et al., 2023a;b; Ouyang et al., 2022) are the foundation of state-of-the-art (SOTA) models for *text* generation (e.g., GPT (Brown, 2020; Radford et al., 2019; Radford, 2018; Achiam et al., 2023)) and *image* generation (e.g., VAR (Tian et al., 2024), LlamaGen (Sun et al., 2024)).

Despite their impressive performance, AR models suffer from slow generation speeds due to their sequential generation process. More specifically, AR models formulate data (e.g., text, images) as a sequence of *tokens* and are trained to predict the conditional probability distribution of *one* token given all previous tokens. This means that AR models can only generate data in a token-by-token

---

[*]Work mostly done during Enshu Liu's internship at Microsoft Research
[†]Project advisor: Zinan Lin

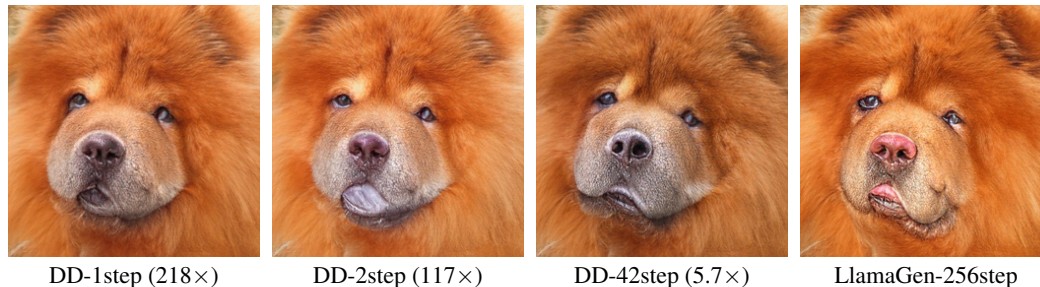

| DD-1step (218×) | DD-2step (117×) | DD-42step (5.7×) | LlamaGen-256step |

Figure 1: **Qualitative comparisons between DD and vanilla LlamaGen Sun et al. (2024) on ImageNet 256×256.** We show that the generated images of DD have small quality loss compared to the pre-trained AR model, while achieving ≥200× speedup. More examples are in App. G.

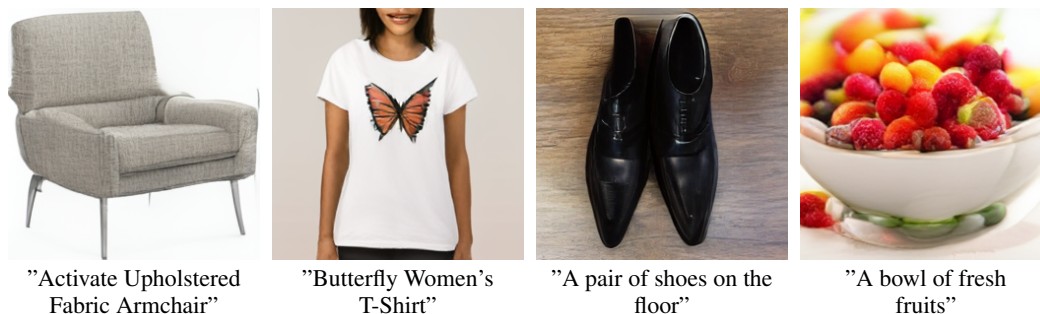

| "Activate Upholstered Fabric Armchair" | "Butterfly Women's T-Shirt" | "A pair of shoes on the floor" | "A bowl of fresh fruits" |

Figure 2: **Qualitative results of DD-2step on text-to-image task.** The model is distilled from LlamaGen model with prompts from LAION-COCO dataset. The speedup is around **93** × compared to the teacher model. More examples are in App. G.

fashion, which is slow. For example, LlamaGen-343M-256×256 requires 256 steps (∼5 seconds[1]) to generate one 256×256 image.

In this paper, we ask the ambitious question:

*Can a pre-trained AR model be adapted to generate data in a few (e.g., one or two) steps?*

If successful, this would greatly benefit both the model developers (e.g., reducing the testing and deployment cost) and the end users (e.g., reducing the latency).

Apparently, this problem is challenging. While speeding up AR models by *generating multiple tokens at a time* (Fig. 4) is extensively studied in literature (Ning et al., 2024a; Liu et al., 2024; Jin et al., 2024; Stern et al., 2018; Kou et al., 2024; Gloeckle et al., 2024; Cai et al., 2024; Santilli et al., 2023; Chang et al., 2022; Li et al., 2024a), *none of these works were able to generate the entire sample (i.e., all tokens) in one step*. Indeed, we find that there is a fundamental reason for this limitation (Sec. 3.1). Sampling multiple tokens in parallel (given previous tokens) would have to assume that these tokens are *conditionally independent with each other*, which is incorrect in practice. As an extreme, generating *all* tokens in *one* step (i.e., assuming that all tokens are independent) would completely destroy the characteristics in data (Sec. 3.1). This insight suggest that few-step AR generation requires a fundamentally different approach.

In this paper, we introduce Distilled Decoding (DD) (Fig. 4), a novel method for distilling a pre-trained AR model for few-step sampling. In each AR generation step, we use flow matching (FM) (Liu et al., 2022; Lipman et al., 2022) to transform a random noisy token, sampled from an isotropic Gaussian distribution, into the generated token. FM ensures that the generated token's distribution aligns with that of the AR model. However, this generation process would be even slower than vanilla AR due to the added FM overhead. The actual benefit is that, the mapping between noisy and generated token is *determinstic*. Therefore, we can train a model that directly distills the mapping between *the entire sequence of noisy tokens* and *the generated tokens*. This enables *one-step generation* by inputting the sequence of noisy tokens into the distilled model. Moreover, this deep synergy between AR and FM allows for *flexible use of additional generation steps to improve data quality without changing the model* (see Sec. 3.3). Note that the training of DD does *not* need the training data of the original AR model. This makes DD more practical as training data is often not released especially for SOTA LLMs.

---

[1]Measured on one NVIDIA A100-80G GPU.

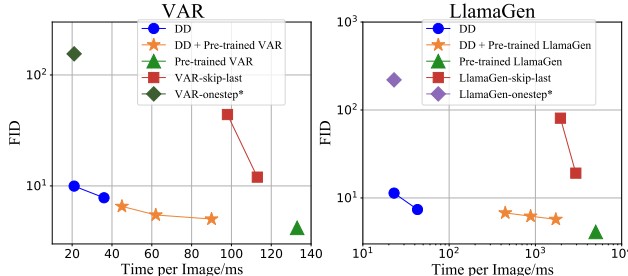

Figure 3: Comparison of `DD` models, pre-trained models, and other acceleration methods for pre-trained models. `DD` achieves significant speedup compared to pre-trained models with comparable performance. In contrast, other methods' performance degrades quickly as inference time decreases.

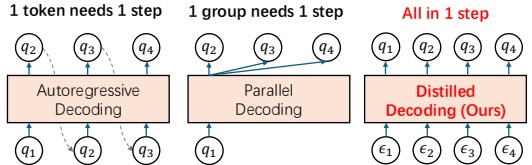

Figure 4: High-level comparison between our `Distilled Decoding` (`DD`) and prior work. To generate a sequence of tokens $q_i$: (a) the vanilla AR model generates token-by-token, thus being slow; (b) parallel decoding generates multiple tokens in parallel (Sec. 4.1), which fundamentally cannot match the generated distribution of the original AR model with one-step generation (see Sec. 3.1); (c) our `DD` maps noise tokens $\epsilon_i$ from Gaussian distribution to the whole sequence of generated tokens directly *in one step* and it is guaranteed that (in the optimal case) the distribution of generated tokens matches that of the original AR model.

As the first work in this series, we focus on *image AR models*. We validate the effectiveness of `DD` on the latest and SOTA image AR models: VAR (Tian et al., 2024) and LlamaGen (Sun et al., 2024). Our key contributions are:

- We identify the fundamental limitations of existing methods that prevent them from achieving few-step sampling in AR models.
- We introduce `DD` to distill pre-trained AR models for few-step sampling.
- For the first time, we demonstrate the feasibility of 1-step sampling with SOTA image AR models. On ImageNet, `DD` reduces the sampling of VAR from 10 steps to 1-step (6.3× speed-up) with an acceptable increase in FID from 4.19 to 9.96; for LlamaGen, `DD` cuts sampling from 256 steps to 1 (217.8× speed-up) with a comparable FID increase from 4.11 to 11.35. In both cases, baseline methods completely fail on 1 step generation and achieve FID >100. For *text-to-image* generation with LlamaGen on LAION-COCO dataset, `DD` is able to reduce the sampling steps from 256 to 2 (92.9× speed-up) with minimal FID increase from 25.70 to 28.95. `DD` also supports more steps for better image quality (Fig. 3). See Figs. 1 and 2 for visualization.

This work challenges the assumption that AR models must be slow. We hope it paves the way for efficient AR models and inspires research into one-step AR models in other areas, such as text generation, where the task is more challenging due to the higher number of steps.

## 2 PRELIMINARIES

In this section, we introduce the preliminaries required to understand `DD`.

### 2.1 AR MODELS

Given a random variable $z$ arranged in a sequence of $n$ tokens $(q_1, \cdots, q_n)$, AR models learn the conditional distribution of tokens given all previous ones: $p(q_i|q_{<i}) = p(q_i|q_{i-1}, q_{i-2}, \cdots, q_1)$. The data likelihood is given by $p(z) = \prod_{i=1}^{n} p(q_i|q_{<i})$. At generation, AR models sample tokens *one by one* using the learned conditional distribution for each token, which is therefore slow.

### 2.2 IMAGE AR MODELS

**Image tokenizer.** To apply AR models to images, we need to represent continuous images as a sequence of discrete tokens. Early works of image AR models operate on quantized pixels (Van den Oord et al., 2016; Chen et al., 2018). Later works propose the vector quantization (VQ) method, utilizing a encoder $\mathcal{E}$, a quantizer $\mathcal{Q}$, and a decoder $\mathcal{D}$ to quantize and reconstruct images. Specifically, after $[\mathcal{E}, \mathcal{Q}, \mathcal{D}]$ is fully trained, the encoder will transform the original image $x \in \mathbb{R}^{3 \times H \times W}$ into

a more compact latent space: $Z = \mathcal{E}(x)$, where $Z = \{z_1, z_2, \cdots, z_{h \times w}\} \in \mathbb{R}^{C \times h \times w}$ consists of $h \times w$ embeddings, each has a dimension of $C$. Then, the quantizer will look up the closest token $q_i$ in the codebook $\mathcal{V} = (c_1, c_2, \cdots, c_V) \in \mathbb{R}^{V \times C}$ for each embedding $z_i$. AR works on the quantized sequence $Z_q = \{q_1, q_2, \cdots, q_{h \times w}\}$. Finally, one can use the decoder to reconstruct the image from the quantized sequence: $\hat{x} = \mathcal{D}(Z_q)$. A distance loss $l(\hat{x}, x)$ is used in training for accurate reconstruction. The VQ scheme is used in many popular image AR models (Li et al., 2023; Chang et al., 2022; Tian et al., 2024; Sun et al., 2024; Li et al., 2024a).

**The order of tokens.** Another important design choice of image AR models is the order of tokens. A typical and straightforward method of ordering tokens is following the raster order of the embeddings $z_i$, such as LlamaGen (Sun et al., 2024). Random order is a more general option (Chang et al., 2022; Li et al., 2023; 2024a). The SOTA method VAR (Tian et al., 2024) proposes a novel method where the tokens are arranged according to the resolution. Specifically, give the latent $Z \in \mathbb{R}^{C \times h \times w}$, VAR down-samples it to a series of resolutions $((h_1, w_1), (h_2, w_2), \cdots, (h_K, w_K))$, where $h_i < h_j, w_i < w_j$ for $i < j$. Then, VAR arranges the augmented tokens from these latents in the order of resolution. Tokens from the same resolution are sampled simultaneously. This more natural and human-aligned next scale prediction approach enables VAR to achieve excellent performance. However, it still needs 10 steps to generate a 256×256 image, making it inefficient.

## 2.3 FLOW MATCHING

Given two random distribution $\pi_0(x), \pi_1(x)$ with $x \in \mathbb{R}^d$, flow matching (FM) (Liu et al., 2022; Lipman et al., 2022) constructs a probability path which connects the two distributions. Setting up a continuous timestep axis and putting the two distribution at $t = 0$ and $t = 1$ respectively, such probability paths can be viewed as an ordinary differential equation (ODE) trajectory. Specifically, flow matching defines a velocity field $V(x, t)$ at any timestep $t$, given by:

$$V(x, t) = \mathbb{E}_{x_0, x_1 \sim \pi_{0,1}(x_0, x_1)}\left(\frac{\partial \varphi(x_0, x_1, t)}{\partial t} | \varphi(x_0, x_1, t) = x\right), \tag{1}$$

where $\pi_{0,1}(x_0, x_1)$ is any joint distribution that satisfies both $\int \pi_{0,1}(x_0, x_1) \mathrm{d}x_0 = \pi_1(x_1)$ and $\int \pi_{0,1}(x_0, x_1) \mathrm{d}x_1 = \pi_0(x_0)$, and $\varphi(x, y, t)$ is a trivariate bijection function for $x$ and $y$, which means that at any give timestep $t$, knowing any two of the $\varphi, x, y$ will determine the third one. $\varphi(x, y, t)$ should also satisfy the boundary conditions: $\varphi(x_0, x_1, 0) = x_0, \varphi(x_0, x_1, 1) = x_1$.

The flow matching ODE has *marginal preserving* property. Given $x_0 \sim \pi_0$, we can obtain $x_t$ by stepping along the ODE trajectory: $x_t = x_0 + \int_{\tau=0}^t V(x_\tau, \tau) \mathrm{d}\tau$. It can be proved that the distribution of $x_t$ is equal to the distribution of $\varphi(x_0, x_1, t)$ (Liu et al., 2022; Lipman et al., 2022): $\pi(x_t) = \pi(\varphi(x_0, x_1, t))$, where $x_0, x_1 \sim \pi_{0,1}(x_0, x_1)$. When we step to $t = 1$, we will get a sample from another distribution $\pi_1$. In practice, $\pi_1$ is usually set as the target distribution we want to sample from, while $\pi_0$ is set as a simple prior distribution, e.g., Gaussian distribution. Then flow matching enables a generative process from the prior to the target.

## 3 METHOD

### 3.1 TRAINING FEW-STEP AR MODELS IS NON-TRIVIAL

In this section, we explain why few-step generation is challenging for AR models and why existing approaches fundamentally cannot achieve it. To finish the sampling process of the whole token sequence $(q_1, \cdots, q_n)$ in as few steps as possible, each step should generate as many tokens as possible. Assume our goal is to generate a set of next tokens $(q_{k+1}, \cdots, q_m)$ based on a subsequence $(q_1, \cdots, q_k)$ as prefix in one step, there are several straightforward ideas to solve this problem. We discuss each method and the reason why they do not work as below:

**(1) Train a neural network with the prefix sequence as input and output the subsequent tokens in one run**. Actually, the information in the prefix $(q_0, \cdots, q_k)$ is not enough to *deterministically* specify the $(q_{k+1}, \cdots, q_m)$, because there are many feasible possibilities for the subsequent tokens that can fit the prefix, leaving a one-to-many mapping for the neural network to learn. For example, we can mask the face portion of a portrait and then select another face from many choices to replace it. In this case, the neural network can only learn an average of all possibilities and output blurry image tokens, like the one in MAE (He et al., 2022). At the stage where the prefix token is insufficient to constrain the subsequent tokens (e.g., at the beginning of the AR generation process), feasible choices are even much more, leading to poor generation quality. A further potential solution is to select the choice with the highest probability as the output target of the network (e.g., Song et al. (2021)). However, such a method will destroy the distribution completely by collapsing into the most likely mode, which is impractical for image generation.

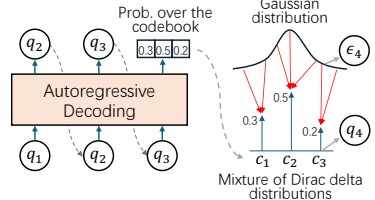

Figure 5: AR flow matching. Given all previous tokens, the teacher AR model gives a probability vector for the next token, which defines a mixture of Dirac delta distributions over all tokens in the codebook. We then construct a deterministic mapping between the Gaussian distribution and the Dirac delta distribution with flow matching. The next noise token $\epsilon_4$ is sampled from the Gaussian distribution, and its corresponding token in the codebook becomes the next token $q_4$.

**(2) Model the distribution of all subsequent tokens.** The neural network can be trained to output the probability of each subsequent token given the prefix as the input. This approach avoids the challenges caused by the one-to-many mapping and is used in mask-based AR models (Chang et al., 2022; Li et al., 2023; 2024a) and VAR (Tian et al., 2024). In this case, the sampling processes of each token are independent from each other, introducing a gap between the modeled distribution $\prod_{i=k+1}^{m} p(q_i|q_k, \cdots, q_1)$ and the ground truth $p(q_m, \cdots, q_{k+1}|q_k, \cdots, q_1)$. For image AR models, the gap is acceptable when $m - k$ is small. However, when the number of new tokens is large, the gap will increase exponentially, making the performance much worse. Consider the case where the model is trained to learn to sample all tokens in one run (which is our goal). In this case, the cross-entropy loss of every token is calculated separately. The final objective can be written as

$$\mathcal{L} = \frac{1}{N}\sum_{i=1}^{N}\frac{1}{n}\sum_{j=1}^{n}\sum_{k=1}^{V}p_{ijk}\log\hat{p}_{\theta jk}, \tag{2}$$

where $V$ is the codebook size, $N$ is the dataset size, $p_{ijk}$ is a one-hot vector along the third dimension given any $(i, j)$, indicating the ground-truth probability distribution, and $\hat{p}_{\theta jk}$ is the modeled distribution. In the following proposition, we give the optimal solution of $\hat{p}_{\theta jk}$.

**Proposition 3.1.** *The optimal solution for Eq. (2) is* $\hat{p}_{\theta^* jk} = \frac{\sum_{i=1}^{N}p_{ijk}}{N}$

This solution equals to occurrence frequency of the $k$-th token at the $j$-th position within the dataset. The proof can be found in App. A. Based on this proposition, consider a toy case where the dataset only contains 2 data samples: $\mathcal{D} = \{(0, 0), (1, 1)\}$. It is easy to find the optimal one-step sampling distribution is a uniform distribution among $\{(0, 0), (1, 1), (0, 1), (1, 0)\}$, which is incorrect. It indicates that the widely used method which predicts the next *group* of tokens is fundamentally impossible to apply for few-step generation. We will see such experimental evidence in Sec. 5.

Additionally, attempting to model the joint distribution by outputting the probability distribution over all possible next $m - k$ tokens is also impractical because of the large number $V^{m-k}$ of possible values, where $V$ is the codebook size, which is typically several thousands or even tens of thousands.

## 3.2 CONSTRUCTING DETERMINISTIC TRAJECTORIES FROM GAUSSIAN TO DATA THROUGH AR FLOW MATCHING

As discussed in section Sec. 3.1, training a model to generate all the tokens in a single run is impossible due to the issue of one-to-many mapping. Therefore, constructing a one-to-one mapping is the key for training a model to generate more tokens simultaneously. The process is illustrated in Fig. 5.

**Construct a mapping for a single token.** Consider the sampling process of a single token $q_i \in \mathbb{R}^C$ given $(q_1, \cdots, q_{i-1})$. Inspired by the knowledge distillation method for diffusion model (Luhman & Luhman, 2021; Salimans & Ho, 2022; Song et al., 2023; Song & Dhariwal, 2023; Kim et al., 2023) which map noise to data, we propose to use **noise token** as additional information to determine this single token. Specifically, we sample noise token from a prior distribution and map it to the generated token. We hope that **(1)** every noise token will be transferred to a deterministic data token in the codebook, and **(2)** the distribution of generated token equals to $p_\theta(q_i|q_{i-1}, \cdots, q_0)$ given by the pre-trained AR model. We propose to use flow matching as the desired mapping for the single token. We operate on the continuous embedding space $\mathbb{E}^C$ of each token given by the codebook $\mathcal{V} = (c_1, \cdots, c_v) \in \mathbb{R}^{V \times C}$. Since the distribution of the next token is a discrete probability distribution among all tokens in the codebook: $\mathbb{P}(q_i = c_j) = p_j$, where $p_j \geq 0, \sum_{j=1}^{V} p_j = 1$, it can also be viewed as a weighted sum of point distributions in the continuous embedding space: $p(q_i) = \sum_{j=1}^{V} p_j \delta(q_i - c_j)$, where $\delta(\cdot)$ is the Dirac function. Denoting $\pi_t$ as the marginal distribution at timestep $t$, we set this weighted sum of point distributions as the target distribution $\pi_1$ and apply a standard Gaussian distribution as the source distribution $\pi_0$. We further choose the linear interpolation used in Liu et al. (2022) as the perturb function: $\varphi(z_0, z_1, t) = (1-t)z_0 + tz_1$. Then

the velocity field in the flow matching framework is given as:

$$V(x,t) = \frac{\sum_{j=1}^{V} p_j(c_j - x)e^{\frac{(x-tc_j)^2}{(1-t)^2}}}{(1-t)\sum_{j=1}^{V} p_j e^{\frac{(x-tc_j)^2}{(1-t)^2}}} \tag{3}$$

In this way, we construct a mapping **from a noise token embedding** $\epsilon_i \in \mathbb{R}^C$ **to the generated token embedding** $q_i$ with the ODE $dx = V(x,t)dt$ while keeping the distribution of $q_i$ unchanged. We denote this mapping as

$$q_i = FM(\epsilon_i, p_\theta(\cdot|q_{<i})) \tag{4}$$

In practice, we use numerical solvers (Lu et al., 2022a;b) to solve the ODE process, so $q_i$ will not have an exact match in the codebook. To tackle that, we use the token closest to $q_i$ in the codebook.

**Construct the whole trajectory along the AR generation process.** After constructing a mapping from $(q_1, \cdots, q_{i-1}, \epsilon_i)$ to $(q_1, \cdots, q_{i-1}, q_i)$, we can extend such mapping to situations with arbitrary length of subsequent noise tokens $(q_1, \cdots, q_{i-1}, \epsilon_i, \cdots, \epsilon_m)$ via an iterative way. After generating the current token $q_i$ from noise $\epsilon_i$, we can update the sequence with $(q_1, \cdots, q_i, \epsilon_{i+1}, \cdots, \epsilon_n)$ and get the conditional probability $p(q_{i+1}|q_i, \cdots, q_1)$. Then the same method can be applied to transfer the next noise token $\epsilon_{i+1}$ to data token $q_{i+1}$, until all the subsequent noise tokens are mapped to the corresponding generated data. This process imposes no constraints on the length of the prefix, so we can start from a pure noise sequence $(\epsilon_1, \cdots, \epsilon_n)$. In this way, we construct an AR trajectory $\{X_i\}_{i=1}^{n+1}$ from pure noise to the final data using flow matching, given as:

$$X_i = (q_1, \cdots, q_{i-1}, \epsilon_i, \cdots, \epsilon_n) \tag{5}$$

The recurrence relation is given by Eq. (4).

### 3.3 DISTILLING AR ALONG THE TRAJECTORY

The AR trajectory transfer the pure noise sequence progressively to the final data sequence, thus is suitable for a neural network to learn. To enable a trade-off between sample quality and sample step, we train the model to predict the final data not only at the beginning of the trajectory but also at intermediate points. Based on this, we first clarify the notation, discuss the model parameterization and training loss, and finally introduce the overall workflow of distillation and sampling.

**Notation.** Suppose $X = (x_1, \cdots, x_n)$ and $Y = (y_1, \cdots, y_n)$ are two arbitrary sequence. We denote $X[: t]$ as a sub-sequence with the first $t$ tokens in $X$ and $X[t + 1 :]$ is the rest part: $X[: t] = (x_1, \cdots, x_t), X[t + 1 :] = (x_{t+1}, \cdots, x_n)$.[2] Additionally, we define the $Concat$ operation as concatenating two sequences: $Concat(X, Y) = (x_1, \cdots, x_n, y_1, \cdots, y_n)$.

**Model parameterization.** The model takes an intermediate value $X_t = (q_1, \cdots, q_{t-1}, \epsilon_t, \cdots, \epsilon_n)$ of the trajectory and the position $t$ as input, and output the final data $X_{n+1} \in \mathbb{R}^{n \times C}$ corresponding to the $X_t$. Suppose we have a neural network $f_\theta$ and denote its output as $f_\theta(X) = (f_{\theta 1}, \cdots, f_{\theta n})$. Since the first $t - 1$ tokens of $X_t$ is already correct tokens, they can be directly used as the first $t - 1$ tokens of the predicted results. We thus parameterize the model output as:

$$F_\theta(X_t, t) = Concat(X_t[: t - 1], f_\theta(X_t)[t :]) = (q_1, \cdots, q_{t-1}, f_{\theta t}, \cdots, f_{\theta n}) \tag{6}$$

**Training loss.** We train the model by minimizing the distance between its output and the target data sequence, which gives the following loss:

$$\mathcal{L} = \mathbb{E}[\lambda(t)d(F_\theta(X_t, t), X_{n+1})] \tag{7}$$

Denote $\{t_1, \ldots, t_L\}$ as the set from which timestep $t$ can be sampled, where $t_i \in [1, n]$ and $t_1 = 1$, and $X_t = Concat(X_n[: t - 1], X_1[t :])$, where $X_n$ is the final generated data by the pre-trained AR model corresponding to $X_1$. The expectation of Eq. (7) is taken with respect to $t \sim \text{Uniform}\{t_i\}_{i=1}^{L}$ and $X_1 \sim \mathcal{N}(0, \boldsymbol{I})$. $\lambda(\cdot) \neq 0$ is a step-wise weighted function. $d(\cdot, \cdot)$ is any distance function that satisfies $d(x, y) \geq 0$ and $d(x, y) = 0$ if and only if $x = y$.

**Overall distillation workflow (Fig. 6).** Our overall workflow consists of two parts: generating the training set and training the model. First, we iteratively draw noises from standard Gaussian distribution and calculate the deterministic generated data through the method in Sec. 3.2, to construct a dataset of noise-data pairs (see Alg. 1). Then we train the model with Eq. (7) using the dataset (see Alg. 2). In this case, the noise-data pair $(X_1, X_n)$ is directly drawn from the dataset.

---

[2]Note that this is different from Python indexing.

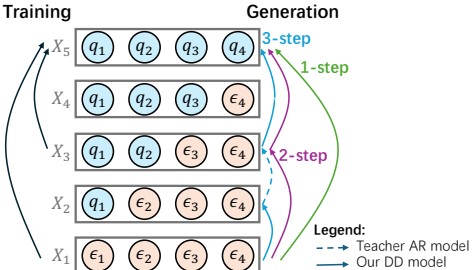

Figure 6: The training and generation workflow of DD. Given $X_1$ with noise tokens $\epsilon_i$, the whole trajectory $X_1, \cdots, X_5$ consists of data tokens $q_i$ and noise tokens $\epsilon_i$ is *uniquely* determined (Sec. 3.2). Assuming the timesteps are set to $\{t_1 = 1, t_2 = 3\}$. During training (Sec. 3.3), we train DD model to reconstruct $X_5$ given $X_1$ or $X_3$ as input. The DD will then have the capability of jumping from $X_1$ and $X_3$ to *any point in the later trajectory* (e.g., $X_1$ to any of $\{X_2, \cdots, X_5\}$). During generation (Sec. 3.3), we can either do 1-step ($X_1 \to X_5$) or 2-step generation ($X_1 \to X_3 \to X_5$). Additionally, we can do generation with more steps by incorporating the teacher AR model in part of the generation process, such as 3-step generation $X_1 \to X_2 \to X_3 \to X_5$ where $X_2 \to X_3$ utilizes the AR model and other steps use the DD model.

**Generation (Fig. 6).** With *one trained DD model* using the above workflow, we can *flexibly adjust the trade-off between sample quality and the number of generation timesteps during inference* (Alg. 3). Specifically, any subset of the training timesteps $\{t_{k_1}, \cdots, t_{k_l}\}$ where $t_{k_1} = 1$ can be chosen as a timestep path from noise to data. We can choose only the first timestep $\{t_1\}$ to do one-step generation, or leverage more timesteps to improve sample quality. Utilizing the AR property of the trajectory, jumping from step $t_{k_i}$ to $t_{k_{i+1}}$ can be implemented by predicting all tokens at timestep $t_{k_i}$ and only keeping the ones before $t_{k_{i+1}}$.

Additionally, we can utilize the pre-trained AR model to achieve more flexibility in the trade-off between sample quality and timestep. Specifically, we can use the pre-trained AR model to generate from any position along the sampling process until reaching the next trained timestep for few-step sampling. Alg. 4 is an example of $2 + t_{k_2} - t_s$ step sampling with the pre-trained model inserted into the original two-step sampling process. We use the DD model on the first step, re-generate the last $t_{k_2} - t_s$ tokens with the pre-trained AR model, and then apply the DD model for the second step.

**Discussions.** As a deep synergy between AR and flow matching, *DD possesses interesting properties that none of the flow matching (which is closely related to diffusion models) and AR have*. Compared to AR, DD has the capability of adjusting the number of generation steps. Compared to flow matching (and diffusion models), DD uses AR to construct the trajectories, and therefore, DD has an easy way to jump back to any point in the trajectory by replacing the last few generated tokens with the original noise tokens, which we utilize in the sampling procedure above. This property enables DD to keep the trajectory unchanged when sampling with multi-steps, in contrast to other intermediate-to-final distillation methods for diffusion models (Song et al., 2023; Song & Dhariwal, 2023) which cannot maintain the trajectory during multi-step sampling.

Another point to mention is that the DD does *not* require the training data of the original AR model, as opposed to some related work (Li et al., 2024b; Song et al., 2023). This makes DD more practical in cases where the training data of the pre-trained AR models is *not* released.

## 4 RELATED WORK

### 4.1 DECREASING THE NUMBER OF SAMPLING STEPS OF AR MODELS

**Image generation.** **(1)** *Mask-based image AR methods* (Chang et al., 2022; Li et al., 2023; 2024a) adopt a random masking ratio when training, allowing them to directly perform a trade-off between performance and number of steps. Specifically, these models are trained to predict an arbitrary number of tokens given some other arbitrary number of tokens. By increasing the number of tokens generated in each step, they can sample faster at the cost of loss in generation quality. However, the fewest number of generation steps evaluated in these works is only 8 and we show in Sec. 3.1 that they fundamentally cannot support very few steps (e.g., one or two steps). **(2)** For *causal-transformer-based methods*, how to reduce their number of steps is still unknown.

**Text generation (large language models).** There are already many works focusing on speeding up the generation of LLMs (Zhou et al., 2024). **(1)** *Speculative decoding* first generates a draft of multiple following tokens and then applies the target LLM to verify these tokens in parallel. The draft tokens can either be generated (a) sequentially by another (small) AR model (Chen et al., 2023; Leviathan et al., 2023; Li et al., 2024b) or (b) in parallel by specially trained models or

heads (Cai et al., 2024; Gloeckle et al., 2024; Xia et al., 2023). Method (a) does not reduce the number of generation steps and therefore is irrelevant to our few-step generation goal. Method (b) fundamentally cannot match the distribution of the target AR model (Sec. 3.1). As a result, even when generating a small number of tokens at a time, they rely on a verification step to ensure correctness, let alone one-step generation. **(2)** In contrast to *token-level parallelism* exploited in speculative decoding, *content-based parallelism* is another paradigm that exploit the weak relation between different parts of the generated content (Ning et al., 2024a; Liu et al., 2024; Jin et al., 2024). These methods prompt or train the LLM to generate an answer outline, enabling the LLM to generate independent points in the outline in parallel. However, these methods change the output distribution of the original AR model. **(3)** Conceptually similar to our methods, CLLMs (Kou et al., 2024) utilize *Jacobi decoding* trajectory (Song et al., 2021; Santilli et al., 2023) and train a student model to conduct one-step sampling along the trajectory. However, CLLMs only support greedy decoding, which deviates from the (non-deterministic) generated distribution of the original AR model.

In contrast to all these methods, *DD supports one-step sampling while theoretically having the potential to match the output distribution of the original AR model.*

### 4.2 DIFFUSION DISTILLATION

Similar to AR models, diffusion models (Sohl-Dickstein et al., 2015; Ho et al., 2020; Song et al., 2020) also suffer from a large number of sampling steps. Knowledge distillation is a promising approach to address this problem and has been widely studied. These methods leverage the trajectory constructed by the pre-trained diffusion model, and train neural networks to directly predict the value multiple steps ahead in the trajectory, rather than predicting just one step ahead as in the pre-trained model. One key difference between various knowledge distillation methods lies in how the starting and ending points of multi-step skipping are selected. The earliest method (Luhman & Luhman, 2021) trains the model to skip the entire trajectory (i.e., predicting the final data directly from the initial Gaussian noise), enabling one-step sampling. PD (Salimans & Ho, 2022) progressively merges two adjacent steps into one, which makes the optimization task smoother and provides a trade-off between steps and quality. CM (Song et al., 2023; Song & Dhariwal, 2023) skips from any intermediate points along the trajectory to the final data. CTM (Kim et al., 2023) proposes a more general framework which enables the transfer between any two points in the trajectory.

Unlike diffusion models, AR models do not come with deterministic trajectories, and therefore, doing distillation on AR is not straightforward. One of our key innovations is the construction of deterministic trajectories out from a pre-trained AR (Sec. 3.2), thereby making knowledge distillation for AR possible. In this paper, we only explored a simple knowledge distillation paradigm, and we hope that this work opens the door for more knowledge distillation paradigms for AR in the future.

See App. F for more discussions on related works (Li et al., 2024a; Yin et al., 2024)

## 5 EXPERIMENTS

In this section, we use DD to accelerate pre-trained image AR models and demonstrate its effectiveness. More details are in App. D. Ablation studies are in App. E.2.

### 5.1 SETUP

**Training.** For pre-trained AR models, we choose VAR (Tian et al., 2024) and LlamaGen (Sun et al., 2024) for the following reasons: **(1)** Both methods are recently released, very popular, and have achieved excellent performance (e.g., LlamaGen claims to beat diffusion models on image generation). **(2)** Their token sequences are defined very differently (Sec. 2.2): VAR uses images of different resolutions to construct the sequence, while LlamaGen adopts traditional method where tokens are latent pixels and arranged in raster order. We want to test the universality of DD across different token sequence definitions. **(3)** Their number of generation steps vary significantly: VAR has 10 steps while LlamaGen uses 256 steps. We want to test whether DD is able to achieve few-step generation no matter whether the AR model has a small or a large number of steps. **(4)** They both provide different sizes of pre-trained models for us to study how DD scales with model sizes.

Following these works, we choose ImageNet 256×256 as the main benchmark. We set the number of available timesteps as 2 and use $\{1, 6\}$, $\{1, 81\}$ as $\{t_{k_1}, t_{k_2}\}$ for VAR, LlamaGen, respectively. Results show that DD can compress both models to **1 or 2** steps with comparable quality.

We further evaluate DD on text-to-image models to assess its capability in handling large-scale tasks. We select the stage-1 text-to-image model from LlamaGen as our base model, which uses 256 steps

Table 1: Generative performance on class-conditional ImageNet-256. "#Step" indicates the number of model inference to generate one image. "Time" is the wall-time of generating one image in the steady state. Results with $\dagger$ are from Tian et al. (2024). See Tab. 4 for results on more model sizes.

| Type | Model | FID↓ | IS↑ | Pre↑ | Rec↑ | #Para | #Step | Time |
|---|---|---|---|---|---|---|---|---|
| GAN$^\dagger$ | StyleGan-XL (Sauer et al., 2022) | 2.30 | 265.1 | 0.78 | 0.53 | 166M | 1 | 0.3 |
| Diff.$^\dagger$ | ADM (Dhariwal & Nichol, 2021) | 10.94 | 101.0 | 0.69 | 0.63 | 554M | 250 | 168 |
| Diff.$^\dagger$ | LDM-4-G (Rombach et al., 2022) | 3.60 | 247.7 | – | – | 400M | 250 | – |
| Diff.$^\dagger$ | DiT-L/2 (Peebles & Xie, 2023) | 5.02 | 167.2 | 0.75 | 0.57 | 458M | 250 | 31 |
| Diff.$^\dagger$ | L-DiT-7B (Peebles & Xie, 2023) | 2.28 | 316.2 | 0.83 | 0.58 | 7.0B | 250 | >45 |
| Mask.$^\dagger$ | MaskGIT (Chang et al., 2022) | 6.18 | 182.1 | 0.80 | 0.51 | 227M | 8 | 0.5 |
| AR$^\dagger$ | VQVAE-2$^\dagger$ (Razavi et al., 2019) | 31.11 | ∼45 | 0.36 | 0.57 | 13.5B | 5120 | – |
| AR$^\dagger$ | VQGAN (Esser et al., 2021) | 15.78 | 74.3 | – | – | 1.4B | 256 | 24 |
| AR$^\dagger$ | ViTVQ (Yu et al., 2021) | 4.17 | 175.1 | – | – | 1.7B | 1024 | >24 |
| AR$^\dagger$ | RQTran. (Lee et al., 2022) | 7.55 | 134.0 | – | – | 3.8B | 68 | 21 |
| AR | VAR-d16 (Tian et al., 2024) | 4.19 | 230.2 | 0.84 | 0.48 | 310M | 10 | 0.133 |
| AR | LlamaGen-L (Sun et al., 2024) | 4.11 | 283.5 | 0.85 | 0.48 | 343M | 256 | 5.01 |
| Baseline | VAR-*skip-1* | 9.52 | 178.9 | 0.68 | 0.54 | 310M | 9 | 0.113 |
| Baseline | VAR-*skip-2* | 40.09 | 56.8 | 0.46 | 0.50 | 310M | 8 | 0.098 |
| Baseline | VAR-*onestep** | 157.5 | – | – | – | – | 1 | – |
| Baseline | LlamaGen-*skip-106* | 19.14 | 80.39 | 0.42 | 0.43 | 343M | 150 | 2.94 |
| Baseline | LlamaGen-*skip-156* | 80.72 | 12.13 | 0.17 | 0.20 | 343M | 100 | 1.95 |
| Baseline | LlamaGen-*onestep** | 220.2 | – | – | – | – | 1 | – |
| Ours | VAR-d16-DD | 9.94 | 193.6 | 0.80 | 0.37 | 327M | **1** | **0.021** (6.3×) |
| Ours | VAR-d16-DD | 7.82 | 197.0 | 0.80 | 0.41 | 327M | 2 | 0.036 (3.7×) |
| Ours | LlamaGen-L-DD | 11.35 | 193.6 | 0.81 | 0.30 | 326M | **1** | 0.023 (**217.8×**) |
| Ours | LlamaGen-L-DD | 7.58 | 237.5 | 0.84 | 0.37 | 326M | 2 | 0.043 (116.5×) |

for generation. For evaluation, we generate images based on 5k unseen prompts from the LAION-COCO dataset and report the FID between the generated images and their corresponding real images. All other settings are consistent with those used for the ImageNet dataset.

**Generation.** We apply Alg. 3 with 1 and 2 steps as our main results in Sec. 5.2. We additionally use Alg. 4 in Sec. 5.3 for better quality with more generation steps. We use a series of staring timesteps for the pre-trained model to get a smooth trade-off between generation cost and quality.

**Baselines.** Since there is no existing method for decreasing the generation steps of visual AR with causal transformers (Sec. 4), we design two baselines: **(1)** Directly skipping the last several steps, denoted as *skip-n* where n is the number of skipped steps, and **(2)** predicting the distribution of all tokens in one step with the optimal solution in Prop. 3.1, denoted as *onestep\**. We also compare with the base pre-trained AR models and other generative models for comprehensiveness.

## 5.2 MAIN RESULTS

The main results of DD are shown at Tab. 1. The key takeaways are:

**DD enable few-step generation of AR models.** By comparing the required generation step and time between the pre-trained models (VAR and LlamaGen) and our distilled models (VAR-DD and LlamaGen-DD), we can see that DD compresses the step and accelerates the pre-trained AR by a very impressive ratio (6.3× on VAR and 217.8× on LlamaGen). Notably, DD decreases the generation step and time of LlamaGen by *two orders*.

**Baselines do not work for few-step generation.** From Tab. 1, we can see that DD steadily surpasses the two types of baselines. For *skip* baseline, we find that the performance rapidly declines as the number of skipped steps increases. The *onestep\** indeed does not work as expected in Sec. 3.1, due to the lack of correlation between tokens.

**DD does not sacrifice quality too much.** While achieving significant speedup, DD does not sacrifice the generation quality too much. For VAR, the FID increases of one-step and two-step generation are smaller than 6 and 4, respectively. For LlamaGen which has more original steps, the FID increase is around 3.5 for two-step generation. Note that as DD learns the mapping given by the pre-trained AR, its performance is bounded by the pre-trained model. Such results have already outperformed many other popular methods like ADM (Dhariwal & Nichol, 2021) with much faster generation speed.

**DD scales well with different model sizes.** We experiment with two different sizes of LlamaGen and three different sizes of VAR (Tab. 4). DD always achieves reasonable sample quality across different model sizes. In addition, for each model family, with larger model sizes, DD can achieve better FID. These results suggest that DD works and scales well with different model sizes.

Table 2: Generation quality of involving the pre-trained AR model when sampling. The notation *pre-trained-n-m* means that the pre-trained AR model is used to re-generate the $n$-th to $m-1$-th tokens in the sequence generated by the first step of `DD`.

| Type | Model | FID↓ | IS↑ | Pre↑ | Rec↑ | #Para | #Step | Time |
|------|-------|------|-----|------|------|-------|-------|------|
| AR | VAR (Tian et al., 2024) | 4.19 | 230.2 | 0.84 | 0.48 | 310M | 10 | 0.133 |
| AR | LlamaGen (Sun et al., 2024) | 4.11 | 283.5 | 0.865 | 0.48 | 343M | 256 | 5.01 |
| Ours | VAR-*pre-trained-1-6* | 5.03 | 242.8 | 0.84 | 0.45 | 327M | 6 | 0.090 (1.5×) |
| Ours | VAR-*pre-trained-4-6* | 5.47 | 230.5 | 0.84 | 0.43 | 327M | 4 | 0.062 (2.1×) |
| Ours | VAR-*pre-trained-5-6* | 6.54 | 210.8 | 0.83 | 0.42 | 327M | 3 | **0.045** (2.6×) |
| Ours | LlamaGen-*pre-trained-1-81* | 5.71 | 238.6 | 0.83 | 0.43 | 326M | 81 | 1.725 (2.9×) |
| Ours | LlamaGen-*pre-trained-41-81* | 6.20 | 233.8 | 0.83 | 0.41 | 326M | 42 | 0.880 (5.7×) |
| Ours | LlamaGen-*pre-trained-61-81* | 6.76 | 231.4 | 0.83 | 0.40 | 326M | 22 | 0.447 (**11.2×**) |

Table 3: Generation results of `DD` on text-to-image task.

| Type | Model | FID | #Param | #Step | Time |
|------|-------|-----|--------|-------|------|
| AR | LlamaGen | 25.70 | 775M | 256 | 7.90 |
| Ours | LlamaGen-DD | 36.09 | 756M | **1** | **0.052** (151.9×) |
| Ours | LlamaGen-DD | 28.95 | 756M | 2 | 0.085 (92.9×) |

**`DD` allows users to choose the desired trade-off between quality and speed.** From Tab. 1, we can see that generation with two steps has better quality than generation with one step, indicating that `DD` offers a trade-off between quality and step, which is a property that AR models using a causal transformer (such as VAR and LlamaGen) do not have.

### 5.3 GENERATION INVOLVING THE PRE-TRAINED MODEL

In this section, we test the method in Alg. 4 which utilizes both the distilled model from `DD` and the pre-trained AR model to provide more trade-off points between quality and step. As discussed in Sec. 3.3, we use the pre-trained AR model to re-generate token from $t_s$ to $t_{k_2} - 1$, denoted as *pre-trained-$t_s$-$t_{k_2}$*. Results are shown at Tab. 2. We can see this generation method offers a flexible and smooth trade-off between quality and step. For example, for VAR, by totally replacing the first step with the pre-trained model, `DD` reaches a FID of 5.03 which is very close to the pre-trained model, while achieving 1.5× speedup. For LlamaGen, `DD` achieves FID 6.76 with 11.2× speedup.

### 5.4 TEXT-TO-IMAGE GENERATION

In this section, we report the results of `DD` on text-to-image generation in Tab. 3. We can see that DD achieves performance comparable to the pre-trained model (i.e., 25.70 v.s. 28.95) with only 2 sampling steps. Generated examples are demonstrated in Fig. 2 and App. G.

## 6 LIMITATIONS AND FUTURE WORK

**Training few-step AR models without teachers.** In this work, we distill pre-trained teacher AR models to support few-step sampling. The sample quality is therefore bounded by the pre-trained AR model. It would be interesting to explore if it is possible to eliminate the need for a teacher, which offers more flexibility and potentially can lead to better sample quality. As discussed in Sec. 4.2, the trajectory construction in `DD` opens up the opportunity to apply other diffusion distillation approaches to AR models. One potential direction is to apply the teacher-free consistency training approaches (Song et al., 2023; Song & Dhariwal, 2023) on AR models.

**Applications on LLMs.** Theoretically, `DD` can be directly applied to LLMs. However, different from visual AR models, the codebook of LLMs is much larger in both size and embedding dimensions. Additionally, the sequence length of LLMs may be much longer. Therefore, applying `DD` on LLMs introduces new challenges to be solved.

**Questioning the fundamental trade-off between inference cost and performance.** It was recently believed that for AR models scaling up the inference compute and number of generation steps is important for getting better performance, as exemplified by various LLM prompting techniques (Wei et al., 2022; Nori et al., 2023; Ning et al., 2024b; Snell et al., 2024) and the inference scaling law presented in OpenAI o1 (OpenAI, 2024). However, as our experiments demonstrated, it is possible to significantly reduce the inference compute and generation steps without losing too much fidelity, at least for current image AR models. This suggests that while the inference compute is important, current models might be wasting compute in the wrong place. It would be interesting to study (1) where the optimal trade-off between inference cost and performance is, (2) how much current models are away from that optimal trade-off, and (3) how to modify current models or design new models to reach that optimal trade-off.

ACKNOWLEDGEMENT

The authors would like to thank Sergey Yekhanin and Arturs Backurs of Microsoft Research for their support and suggestions of this work, Qian Chen for his help with experiments, and the anonymous reviewers for their valuable feedback and suggestions.

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

## A  PROOF OF PROP. 3.1

*Proof.* Since all positions in the sequence are symmetrical, we can just consider one certain position $j$. The loss on this position is

$$\mathcal{L}_j = \frac{1}{N} \sum_{i=1}^{N} \sum_{k=1}^{V} p_{ijk} \log \hat{p}_{\theta jk} \tag{8}$$

Due to the normalization constraints of probability, we have

$$\sum_{k=1}^{V} \hat{p}_{\theta jk} = 1 \tag{9}$$

Considering the *Lagrange Multiplier Method*, the Lagrange function is given as

$$\mathcal{F}(\hat{p}_{\theta j1}, \cdots, \hat{p}_{\theta jV}, \lambda) = \frac{1}{N} \sum_{i=1}^{N} \sum_{k=1}^{V} p_{ijk} \log \hat{p}_{\theta jk} - \lambda(\sum_{k=1}^{V} \hat{p}_{\theta jk} - 1) \tag{10}$$

The optimal $\hat{p}_{\theta jk}$ and the corresponding $\lambda$ satisfy

$$\frac{\partial \mathcal{F}}{\partial \hat{p}_{\theta jk}} = \frac{\sum_{i=1}^{N} p_{ijk}}{N \hat{p}_{\theta jk}} - \lambda = 0, k = 1, \cdots, V \tag{11}$$

$$\frac{\partial \mathcal{F}}{\partial \lambda} = \sum_{k=1}^{V} \hat{p}_{\theta jk} - 1 = 0 \tag{12}$$

By solving the above equations, we can get the final solution

$$\lambda = \frac{\sum_{i=1}^{N} \sum_{k=1}^{V} p_{ijk}}{N} = 1 \tag{13}$$

$$\hat{p}_{\theta^* jk} = \frac{\sum_{i=1}^{N} p_{ijk}}{N} \tag{14}$$

$\square$

## B  ALGORITHM PSEUDOCODE

In this section, we provide the algorithm pseudocode of training data generation (Alg. 1), training (Alg. 2), and sampling (Alg. 3 and Alg. 4).

## C  MODEL ARCHITECTURE DESIGN

Since our method does not alter the AR property nature of the generation task, we can use the same architecture as the pre-trained model while slightly modify several modules as discussed below. Aside from these adjusted modules, all other modules inherit the weights from the teacher model for quicker convergence.

**Algorithm 1** Generate dataset

**Require:**
    $\theta^{\Phi}$: The pre-trained AR model
**Generation Process**
1: $\mathcal{D} \leftarrow \varnothing$
2: **for** $i = 1, \cdots, N$ **do**
3:     $X_1 \leftarrow (\epsilon_1, \cdots, \epsilon_n) \sim \mathcal{N}(0, \boldsymbol{I})$
4:     **for** $t = 1, \cdots, n$ **do**
5:         $q_t \leftarrow FM(\epsilon_t, p_{\theta^{\Phi}}(\cdot|q_{<t}))$
6:     **end for**
7:     $X_n \leftarrow (q_1, \cdot, q_n)$
8:     $\mathcal{D} \leftarrow \mathcal{D} \cup \{(X_1, X_n)\}$
9: **end for**
10: **return** $\mathcal{D}$

**Algorithm 2** Training

**Require:** :Generated dataset with noise-data pairs $\mathcal{D}$, the pre-trained AR model $\theta^{\Phi}$.
**Hyper-parameter** : Available timesteps for training $\{t_1, \cdots, t_L\}$, learning rate $\eta$.
**Training Process**
1: $\theta$ initialized from $\theta^{\Phi}$
2: **for** $i = 1, \cdots, Iter$ **do**
3:     Sample $(X_1, X_n) \sim \mathcal{D}, t \sim \{t_1, \cdots, t_L\}$
4:     $X_t \leftarrow Concat(X_n[: t-1], X_1[t :])$
5:     $\hat{X}_n \leftarrow F_\theta(X_t, t)$
6:     $\mathcal{L} \leftarrow d(\hat{X}_n, X_n)$
7:     $\theta \leftarrow \theta - \eta\nabla_\theta\mathcal{L}$
8: **end for**
9: **return** $\theta$

**Algorithm 3** Sampling

**Require:** : The distilled few-step model $\theta$, sampling timesteps $\{t_{k_1}, \cdots, t_{k_l}\}$.
**Sampling Process**
1: $X_1 \leftarrow (\epsilon_1, \cdots, \epsilon_n) \sim \mathcal{N}(0, \boldsymbol{I}), X \leftarrow X_1$
2: **for** $t$ in $\{t_{k_1}, \cdots, t_{k_l}\}$ **do**
3:     $X \leftarrow Concat(X[: t-1], X_1[t :])$
4:     $X \leftarrow F_\theta(X, t)$
5: **end for**
6: **return** $X$

**Algorithm 4** Sampling with the pre-trained AR model

**Require:** : The distilled few-step model $\theta$, the pre-trained AR model $\theta^{\Phi}$, sampling timesteps $\{t_{k_1} = 1, t_{k_2}\}$, starting timestep of pre-trained model $t_{k_1} < t_s < t_{k_2}$.
**Sampling Process**
1: $X_1 \leftarrow (\epsilon_1, \cdots, \epsilon_n) \sim \mathcal{N}(0, \boldsymbol{I}), X \leftarrow X_1$
2: $X \leftarrow Concat(X[: t_{k_1} - 1], X_1[t_{k_1} :])$
3: $X \leftarrow F_\theta(X, t_{k_1})$
4: $X \leftarrow Concat(X[: t_s - 1], X_1[t_s :])$
5: **for** $t$ in $\{t_s, \cdots, t_{k_2} - 1\}$ **do**
6:     Sample $q_t \sim p_{\theta^{\Phi}}(\cdot|X[: t-1])$
7:     $X[t] \leftarrow q_t$
8: **end for**
9: $X \leftarrow F_\theta(X, t_{k_2})$
10: **return** $X$

**Two final heads for logits and embedding prediction.** The task of the model is to output the token in the codebook (with a shape of $V \times C$) which corresponds to the input noise. There are two approaches to achieve this goal: we can view the problem as a classification problem among all tokens in the codebook, or we can treat it as a regression task aimed at outputting the correct embedding. Therefore, we set two final heads after the transformer backbone. For each token, one outputs $p \in \mathbb{R}^V$ as *the predicted logits* among all tokens in the codebook, and the other outputs $c \in \mathbb{R}^C$ as *the predicted embedding*. For logits output, we use cross entropy as the distance function $d$ in Eq. (7), while we apply LPIPS loss (Zhang et al., 2018) for the embedding prediction. The overall objective is a weighted sum of the two losses. We empirically find that the predicted logits perform well when $t$ is small and much worse when $t$ is large, while the predicted embeddings perform the opposite. Thus we simply set a split point and use the predicted logits for small $t$s and the predicted embeddings for the rest.

**Additional embeddings for noise and data tokens.** Since data tokens and noise tokens are fed in the model simultaneously, and their distribution differs significantly, it is necessary for the network to distinguish between them. Therefore, before the transformer block, We add two different learnable embeddings to the data tokens and noise tokens, respectively. The two embeddings are randomly initialized.

**Positional Embedding.** Actually, the processed sequence of our model is one token longer than the pre-trained AR model. It is because the pre-trained model needs 1 class label token and $n-1$ generated token to obtain the whole sequence, while our model needs 1 class label token and $n$ noise token to generate the whole sequence. Thus, for models who have positional embedding, we increase its length by 1 and randomly initialize this new part.

**Attention Mask.** The attention mask of the pre-trained AR model also can not be used directly due to the mismatch in length. In our case, we let every token see all previous tokens. For VAR, all tokens generated at the same step can see each other as well.

Table 4: Generative performance on class-conditional ImageNet-256. "#Step" indicates the number of model inference to generate one image. "Time" is the wall-time of generating one image in the steady state.

| Type | Model | FID↓ | IS↑ | Pre↑ | Rec↑ | #Para | #Step | Time |
|------|-------|------|-----|------|------|-------|-------|------|
| AR | VAR-d20 (Tian et al., 2024) | 3.35 | 301.4 | 0.84 | 0.51 | 600M | 10 | 0.184 |
| AR | VAR-d24 (Tian et al., 2024) | 2.51 | 312.2 | 0.82 | 0.53 | 1.03B | 10 | 0.251 |
| AR | LlamaGen-B (Sun et al., 2024) | 5.42 | 193.5 | 0.83 | 0.44 | 111M | 256 | 2.93 |
| Ours | VAR-d20-DD | 9.55 | 197.2 | 0.78 | 0.38 | 635M | **1** | 0.027 (6.9×) |
| Ours | VAR-d20-DD | 7.33 | 204.5 | 0.82 | 0.40 | 635M | 2 | 0.047 (3.9×) |
| Ours | VAR-d24-DD | 8.92 | 202.8 | 0.78 | 0.39 | 1.09B | **1** | 0.034 (**7.4×**) |
| Ours | VAR-d24-DD | 6.95 | 222.5 | 0.83 | 0.43 | 1.09B | 2 | 0.059 (4.3×) |
| Ours | LlamaGen-B-DD | 15.50 | 135.4 | 0.76 | 0.26 | 98.3M | **1** | **0.018** (162.7×) |
| Ours | LlamaGen-B-DD | 11.17 | 154.8 | 0.80 | 0.31 | 98.3M | 2 | 0.029 (101.0×) |

# D    EXPERIMENTAL DETAILS

In this section, we introduce more details of the experiments in Sec. 5.

**Dataset Generation.** In the dataset generation phase, we use Alg. 1 to construct the data-noise pairs. For the ImageNet dataset, we generated a total of 1.2M data-noise pairs; for the text-to-image task, we generated a total of 3M data-noise pairs. For the AR models, we set the classifier-free guidance scale to 2.0 for both VAR and LlamaGen, while other settings follow the default configuration of these two works. In the flow matching process, we employ the perturb function from Rectflow (Liu et al., 2022): $\varphi(x_0, x_1, t) = (1 - t)x_0 + tx_1$. We use the DPM-Solver (Lu et al., 2022a;b) to efficiently solve the FM ODE. Specifically, after calculating the velocity model given by Eq. (3), we wrap the model function $V(x, t)$ to get the noise prediction model $\epsilon(x, t)$: $\epsilon(x, t) = x - tV(x, t)$. Then we can directly use the official implementation of DPM-Solver (see https://github.com/LuChengTHU/dpm-solver) to conduct sampling with the noise prediction model and the Rectflow noise schedule. Our configuration includes 10 NFE multistep DPM-Solver++ with an order of 3.

**Training Configuration.** We follow most of the training configuration for the pre-trained AR model. For VAR, we use a batch size of 512, a base learning rate 1e-4 per 256 batch size, and an AdamW optimizer with $\beta_1 = 0.9, \beta_2 = 0.95$. For LlamaGen, all other settings are the same except for the learning rate, which is fixed at 1e-4 and doesn't vary with the batch size. We distill the VAR model for 120 epochs and LlamaGen model for 70 epochs. We additionally use exponential moving average (EMA) with a rate of 0.9999. As discussed in App. C, our model has two types of prediction results, therefore two types of loss. We assign a loss weight of 1.0 for the embedding loss (LPIPS) and a weight of 0.1 for the logits loss (cross entropy) to maintain a similar loss scale for them. For the timestep-wise loss weight, we use a uniform one with $\lambda(t) = 1$.

# E    ADDITIONAL RESULTS

## E.1    RESULTS WITH MORE MODEL SIZES

In this section, we present results with VAR and LlamaGen models of more parameter sizes in Tab. 4.

## E.2    ABLATION STUDY

**Effect of the training iteration.** From Fig. 7, we can see that the few-step FID decreases rapidly in the first few epochs, demonstrating the high adaptability of the pre-trained AR model across different tasks, which contributes to the smaller distilling cost of DD than training AR models from scratch.

**Effect of the intermediate timestep.** From Fig. 7, we find that the convergence speed and performance of different intermediate timesteps are similar, indicating that DD is not sensitive to it.

**Effect of the dataset size.** From Fig. 8, we find that the DD can still work when there is only 0.6M (data, noise) pairs. Additionally, with more (data, noise) pairs, the performance improves. The results demonstrate DD's robustness to limited data and its data scaling ability.

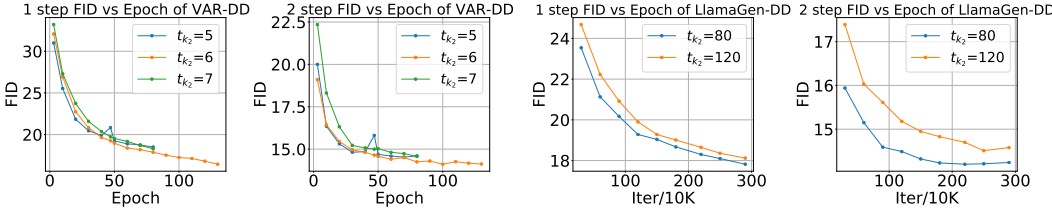

Figure 7: The training curve of FID vs. epoch or iteration for different intermediate timesteps. FIDs are calculated with 5k generated sample.

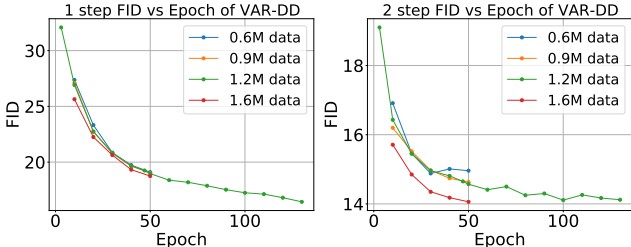

Figure 8: The training curve of FID vs. epoch for different dataset sizes. FIDs are calculated with 5k generated sample.

# F   MORE RELATED WORK DISCUSSIONS

In this section, we provide expanded discussions on two related works: DMD (Yin et al., 2024) and MAR (Li et al., 2024a).

DMD (Yin et al., 2024) uses distribution matching to distill diffusion models into few steps. Specifically, the main idea of DMD is to minimize the distance between the generated distribution and the teacher distribution at all timesteps. Additionally, it constructs data-noise pairs and conducts direct distillation as a regularization loss term to avoid mode collapse. There are several key differences between DD and DMD: **(1) Target:** DMD focuses on diffusion models, whereas DD focuses on AR. These are two different problems; **(2) Technique:** In diffusion models, the diffusion process naturally defines the data-noise pairs, which can be directly used for distillation as in DMD. In contrast, AR models do not have a pre-defined concept of "noise". How to construct noise and the data-noise pair (Sec. 3.2) is one important and unique contribution of our work.

That being said, the distribution matching idea in DMD is very insightful, and could potentially be combined with DD to achieve better approaches for few-step sampling of AR models. For example, from a high-level perspective, the noisy image in DMD is similar to the partial data sequence in AR, while the next token logits given by the pre-trained AR model can be viewed as the score function in DMD. The objective can be set as minimizing the distance between the output of a fake logits prediction network and the pre-trained AR at all timesteps. As long as the next-token conditional distribution of the generated distribution is the same as the distribution given by the pre-trained AR model, the modeled one-step distribution will be correct. There is no requirement for any data-noise pair in this case.

MAR (Li et al., 2024a) proposes to replace the cross-entropy loss with diffusion loss in AR, which shares some similarities with DD at a high level since both works view decoding as a denoising process. The goals of these two works are different though. MAR targets removing the vector quantization in image AR models for better performance, while DD aims to compress the sampling steps of AR, without any modification to the codebook. In this sense, these two works are orthogonal and could potentially be combined to develop a new method that leverages the strengths of both.

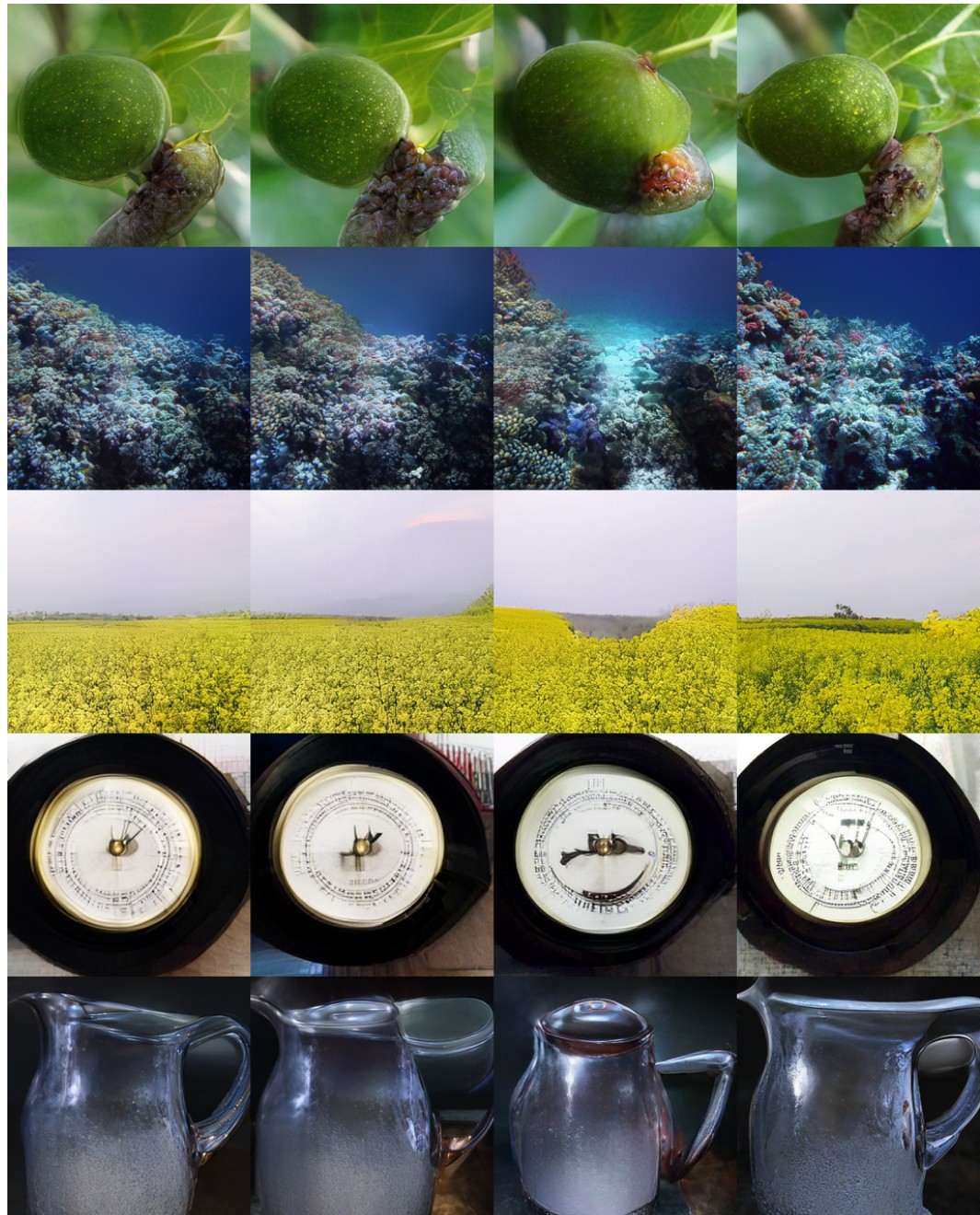

Figure 9: Generation results with VAR model (Tian et al., 2024). From left to right: one-step `DD` model, two-step `DD` model, `DD`-*pre-trained-4-6*, and the pre-trained VAR model.

## G VISUALIZATION

In this section, we demonstrate samples generated by `DD`. Results of label conditional generation are demonstrated in Figs. 9 to 12. Examples of text conditional generation are demonstrated in Figs. 13 to 15, where most texts are taken from the LAION-COCO dataset.

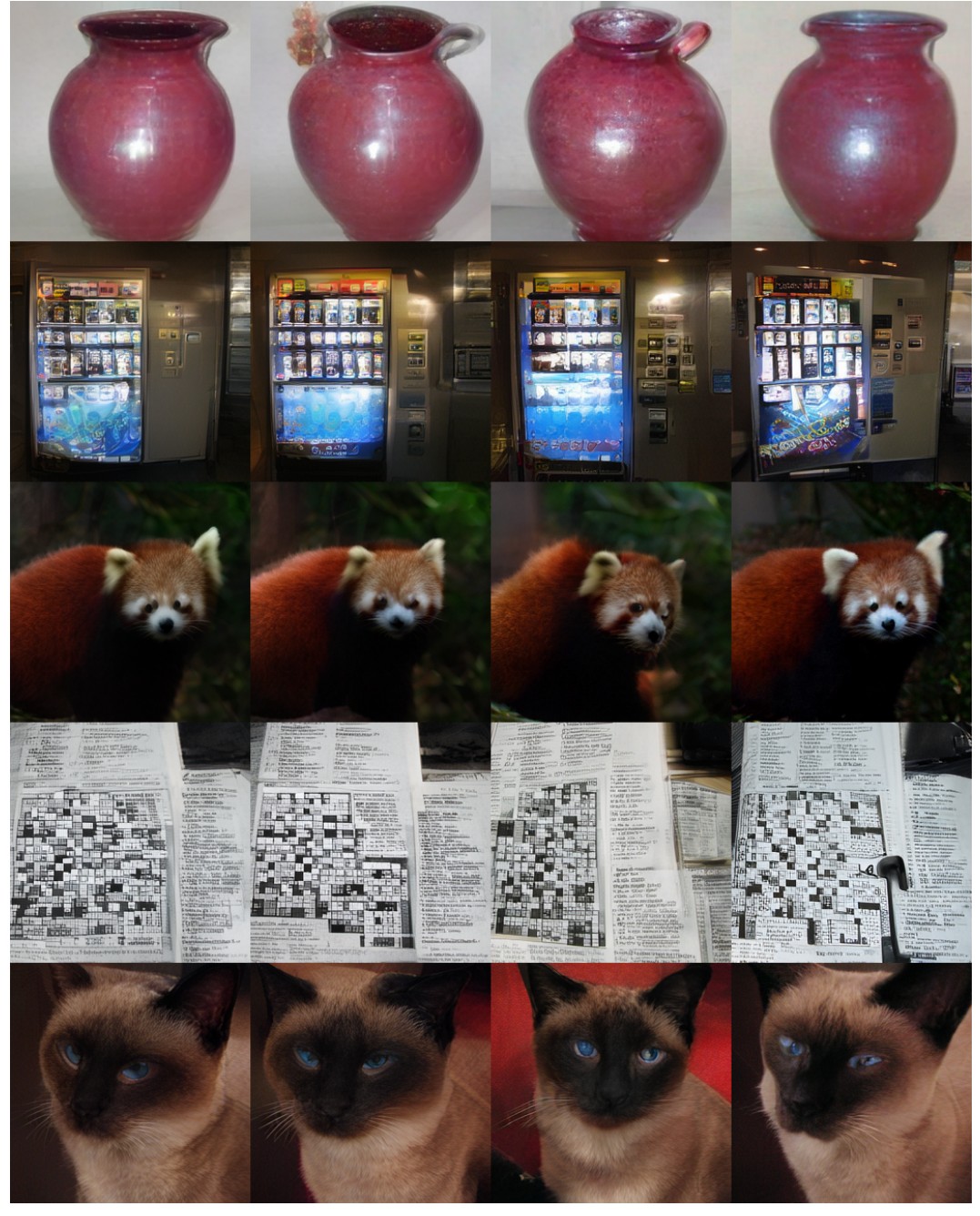

Figure 10: Generation results with VAR model (Tian et al., 2024). From left to right: one-step `DD` model, two-step `DD` model, `DD`-*pre-trained-4-6*, and the pre-trained VAR model.

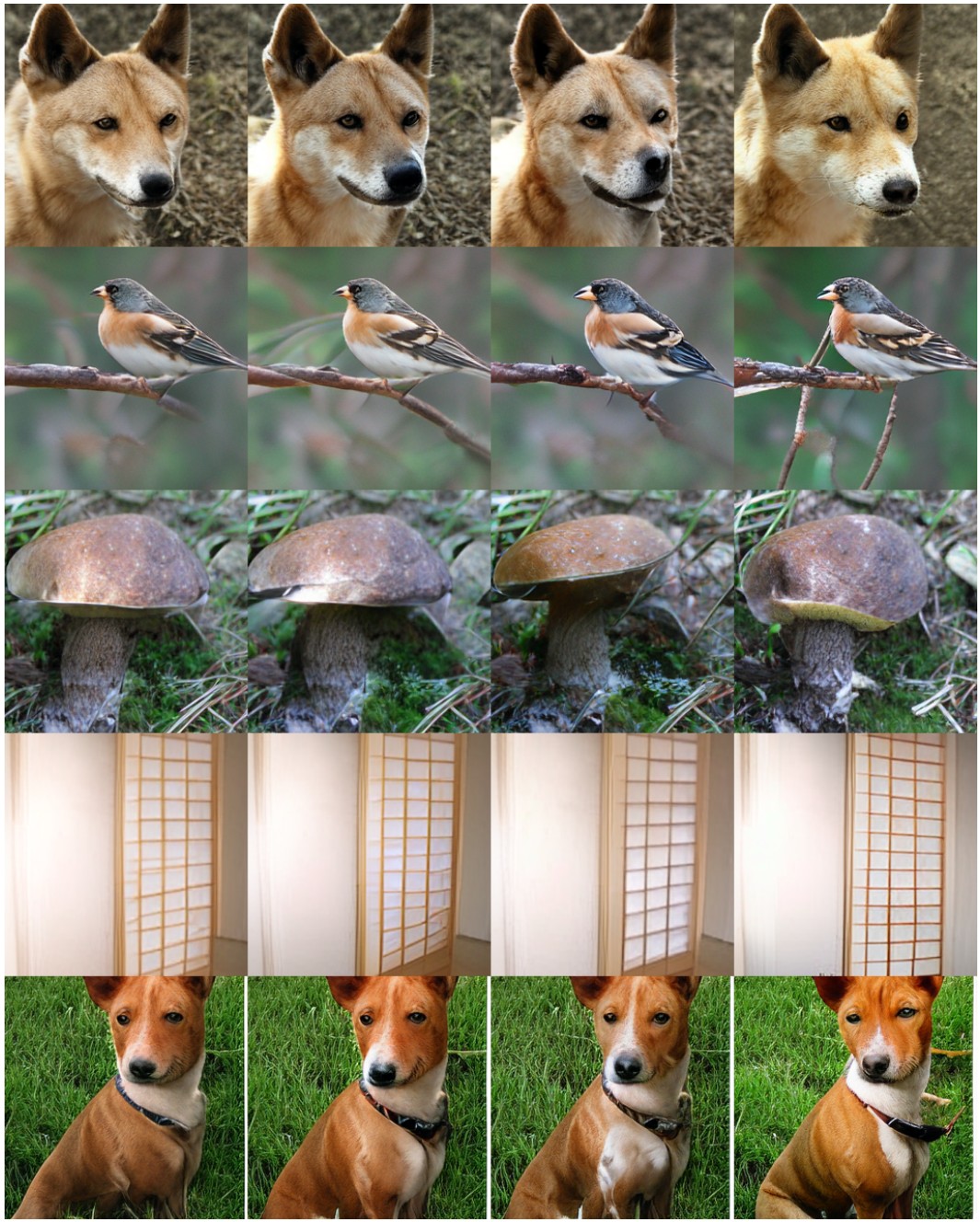

Figure 11: Generation results with LlamaGen model (Sun et al., 2024). From left to right: one-step `DD` model, two-step `DD` model, `DD`-*pre-trained-41-81*, and the pre-trained LlamaGen model.

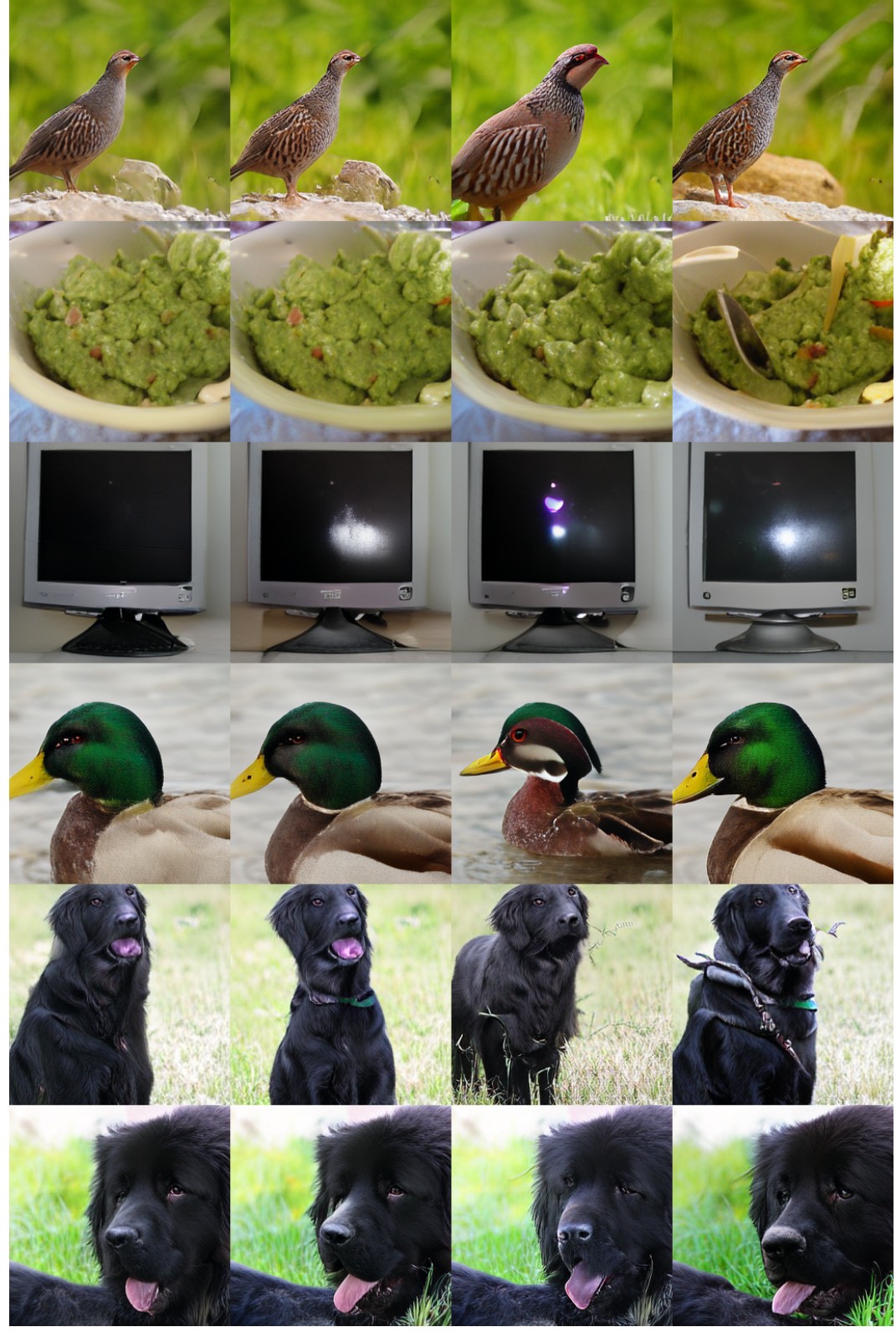

Figure 12: Generation results with LlamaGen model (Sun et al., 2024). From left to right: one-step `DD` model, two-step `DD` model, `DD`-*pre-trained-41-81*, and the pre-trained LlamaGen model.

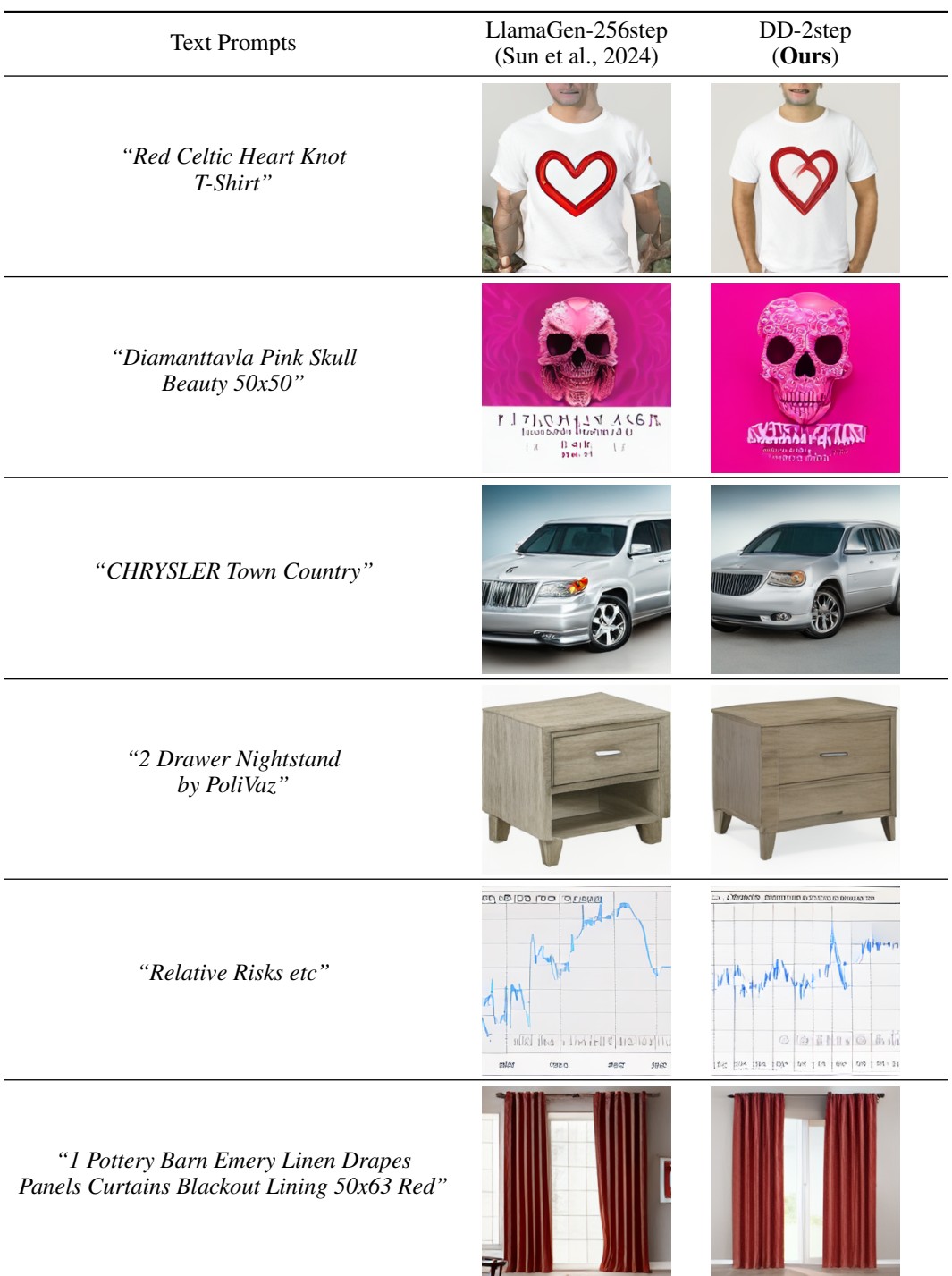

Figure 13: Comparisons of text-to-image results between DD and the pre-trained LlamaGen model (Sun et al., 2024). Prompts without * are taken from the LAION-COCO dataset.

| Text Prompts | LlamaGen-256step
(Sun et al., 2024) | DD-2step
(**Ours**) |
|---|---|---|
| *"WEEKLY or MONTHLY. Soft Touch Genuine Leather Sectional in BARK T-Shirt"* | | |
| *"ToyWatch Men's Plasteramic Diver Watch"* | | |
| *"Wood Wall Behind Tv Ask A Designer Series Mistakes Made To Tv Walls Nesting With Grace"* | | |
| *"Activate Upholstered Fabric Armchair"* | | |
| *"Womens Ladies Studded Ankle Strap Espadrilles Platform Shoes Wedge Sandals Size"* | | |
| *"A pair of shoes on the floor"***\*** | | |

Figure 14: Comparisons of text-to-image results between DD and the pre-trained LlamaGen model Sun et al. (2024). Prompts without **\*** are taken from the LAION-COCO dataset.

| Text Prompts | LlamaGen-256step (Sun et al., 2024) | DD-2step (**Ours**) |
|---|---|---|
| *"Butterfly Women's T-Shirt"* | | |
| *"chanel-cruise-collection-fashion-show-2016-16-colorful-dresses-outfit (58)"* | | |
| *"A tree on a hill"* | | |
| *"Meeting Facility, Holiday Inn Munich-Unterhaching"* | | |
| *"A bridge over a river"* | | |
| *"A bowl of fresh fruits"* | | |

Figure 15: Comparisons of text-to-image results between DD and the pre-trained LlamaGen model Sun et al. (2024). Prompts without * are taken from the LAION-COCO dataset.

