# OpenReview forum: "Distilled Decoding 1: One-step Sampling of Image Auto-regressive Models with Flow Matching"
_ICLR.cc/2025/Conference — ICLR 2025 Poster_

### Official Review · Reviewer_ni7B · 2024-11-01

**Soundness:** 3
**Presentation:** 3
**Contribution:** 3
**Rating:** 6
**Confidence:** 3

**Summary:**

This paper presents a method called Distilled Decoding (DD) that distills autoregressive models into fewer steps. It employs flow matching to establish a deterministic mapping from Gaussian noise to the output distribution of the pretrained autoregressive models. Extensive experiments demonstrate that DD outperforms the baselines, accelerating autoregressive models with minimal performance degradation.

**Strengths:**

1. The paper is well written and easy to follow.
2. The presented idea of compressing AR models with Flow Matching is intersting. Given that autoregressive models tend to be slow, accelerating them is a crucial challenge. The efforts presented in this paper could provide valuable insights for the community.

**Weaknesses:**

1. Lines 201-203 state that modeling the joint distribution of several tokens is impractical due to the vast possible space. However, it seems that the proposed method uses Flow Matching to learn the joint distribution of tokens (the full sequence in the one-step case). This is somewhat confusing—how does the proposed method tackle this issue?
2. The image quality shows significant degradation (as indicated by both FID scores and the figures), raising concerns about the practicality of the proposed method.
3. It would be beneficial to include experiments with text-conditioned LlamaGen to demonstrate that the method can be adapted to more complex scenarios.

**Questions:**

In table 1, why VAR-DD with one step achieves better performance than VAR-DD with two steps?

---

> ### Author Response · Authors · 2024-11-24
> **Response to Reviewer ni7B 1**
>
> Thank you for your great questions! Here are our replies to your concerns and questions:
> - W1: the explanation of why DD can model the joint distributions
>   - We want to clarify that in 201-203, we meant to say that **for vanilla AR** "modeling the joint distribution of several tokens is impractical due to the vast possible space". In contrast, DD makes fundamental changes to the modeling process, which makes it possible. Below, we will explain why vanilla AR fails whereas DD succeeds.
>     - **Vanilla AR** directly outputs the discrete probability distribution for the next token, which is a $V$-dimension because there are $V$ possible values ($V$ is the codebook size). Now, let's consider the joint distribution of $k$ tokens, which is a discrete distribution having $V^{k}$ possible values as every token has $V$ possible values. In this case, the model should output a $V^{k}$-dimension vector in order to model the joint distribution. Since $V$ is often several or tens of thousands, $V^{k}$ will be a very large number even when k is only 2. It is hard for the logit head to learn.
>     - **DD** also models the joint distribution of all tokens but uses a fundamentally different process. Instead of relying on $V^{k}$-dimension output vector to model the distribution of the next $k$ tokens, DD models the distribution *implicitly* by $k$ additional *input* noise tokens. These $k$ input noise tokens, passing through the flow matching we constructed in Section 3.2 and distilled in Section 3.3, deterministically define the next $k$ output tokens. Therefore, DD can model multiple tokens jointly in a more scalable way than the vanilla approach discussed before.
> - W2: the quality shows significant degradation.
>   - We found out that by fixing the incorrect generation configurations and using more training iterations, the sample quality of DD can be substantially improved, getting much closer to the pre-trained model. (1) We first **fix the generation configuration of LlamaGen model** by simply modifying the suboptimal default sampling configuration of the pre-trained model (please see our response to reviewer ziid Q3 for details), which brings an improvement from FID=6.53 to FID=4.11, enabling a higher quality of the teacher distribution. (2) Then, we simply **train both VAR-DD and LlamaGen-DD for more epochs**. These modifications bring significant improvement to the quality of the few-step DD. Specifically, for LlamaGen, the FIDs of 1-step generation and 2-step generation improve from 17.98 and 11.24 to 11.35 and 7.58, and the FID of 81-step generation with pre-trained model also decreases from 10.30 to 5.71, which is very close to the pre-trained model; for VAR, the FID of 1 step generation improves from 10.65 to 9.94, and the FID of 6-step generation improves from 5.90 to 5.03. These results are updated in Table 1 in the revision. Additionally, we update the visualization examples in our revision (please see Fig.3, Fig.8-Fig.11 in our revision), which are of higher quality.
>   - Additionally, DD provides a flexible trade-off between time and quality by involving the teacher model. As shown in Table 2, we can replace the DD model with the pre-trained teacher model in some positions to get better generation quality. Our best results are already close to the pre-trained model (i.e., 4.20 v.s. 5.03 for VAR, 4.11 v.s. 5.71 for LlamaGen), while still having an acceleration ratio of 1.5$\times$ and 2.9$\times$.
>   - We believe that these updated results could address concerns about the sample quality of DD.

---

> ### Author Response · Authors · 2024-11-24
> **Response to Reviewer ni7B 2**
>
> - W3: better to include text-to-image generation.
>   - Thank you for the great suggestions! Following your suggestion, we conduct the following experiments.
>   - **Experiment settings.** We use the official T2I-XL-Stage1 model of LlamaGen (https://huggingface.co/peizesun/llamagen_t2i/resolve/main/t2i_XL_stage1_256.pt) (pre-trained on LAION-COCO) as the pre-trained AR model to be distilled. We generate 1.9M tuples of (text, noise, and data) to train our DD model.
>   - **Results.** Note that T2I is a very difficult task that requires a large amount of data and training costs. Due to the time constraints of the discussion period and many other experiments we added in the rebuttal, we only managed to conduct a small amount of training (around 15 epochs). However, even with such limited training, the DD model already achieves reasonable results. We report the FID scores (evaluated using 5000 generated images) below. As a comparison, the FID of the pre-trained model with 256-step sampling is 25.70. We can see that: (1) the performance of 2-step DD is already close to the pre-trained model (29.90 vs. 25.70), while reducing the number of generation steps by 128 times; (2) The performance of DD is still improving stably at the end of 15 epochs, so we expect that the final performance could be even better. The generated images can be found at https://anonymous.4open.science/r/DD_t2i_ano-F222 .
> | | | | | | | | |
> |-|-|-|-|-|-|-|-|
> | |100k|200k|300k|400k|460k (~15 epochs)| | |
> |1 step|51.66|43.21|41.17|40.05|39.04| | |
> |2 step|56.31|34.12|31.75|30.66|29.90| | |
> | | | | | | | | |
>   - Although the T2I training is not yet complete (and likely will not be finished before the discussion period ends), we believe our current methods and results already represent a solid contribution. **(1)** 1-step generation of image AR models is a challenging problem, even for non-T2I (i.e., label-to-image generation) tasks. We are not aware of, even among concurrent ICLR submissions, any approaches that can do 1-step generation of image AR models. The reason is fundamental--as we discussed in Section 3.1, existing methods based on vanilla AR are fundamentally impossible to achieve good 1-step generation. We believe that DD, as the first method that enables reasonable 1-step generation of AR models, makes non-trivial and solid contributions to the field.  **(2)** On top of that, we already got promising and non-trivial T2I results even with limited training iterations. We believe that the current results have already demonstrated the potential of DD to be applied in such tasks.
>   - We will update the T2I results in the revision once the training is converged.
> - Q1: In table 1, why VAR-DD with one step achieves better performance than VAR-DD with two steps?
>   - Thank you for catching this typo! The results for one-step and two-step VAR-DD were mistakenly flipped. We have corrected this in the revision.

---

> > ### Comment · Reviewer_ni7B · 2024-11-25
> >
> > Thank you for the clarification. I have increased my score to 6.

---

> > > ### Author Response · Authors · 2024-11-25
> > >
> > > Thank you for your positive feedback! We are opening to discussion if you have additional questions.

---

### Official Review · Reviewer_2cqS · 2024-11-02

**Soundness:** 3
**Presentation:** 4
**Contribution:** 3
**Rating:** 8
**Confidence:** 4

**Summary:**

The paper is concisely written with a clear line of thought. The approach of constructing a continuous embedding space in AR and matching it to a Gaussian distribution is particularly interesting. This construction effectively combines the discrete cross-entropy model from AR-based methods with the probability distribution strategies that have proven successful in diffusion models, allowing the concept of consistency distillation from diffusion to be successfully applied within the AR field. The comparisons are comprehensive, and the experiments are well-executed.

**Strengths:**

The paper is concisely written with a clear line of thought. The approach of constructing a continuous embedding space in AR and matching it to a Gaussian distribution is particularly interesting. This construction effectively combines the discrete cross-entropy model from AR-based methods with the probability distribution strategies that have proven successful in diffusion models, allowing the concept of consistency distillation from diffusion to be successfully applied within the AR field. The comparisons are comprehensive, and the experiments are well-executed.

**Weaknesses:**

The paper successfully applies the consistency distillation (CD) technique from diffusion models to the AR field. However, the current results still fall short compared to the pre-trained model.

**Questions:**

My questions lean more toward potential future design possibilities, as the explanations provided in the paper are relatively clear.

Q1: The approach seems very similar to a variant of the Consistency Model (CM) applied to the AR domain, replacing the diffusion model in CM with an AR model. What are the advantages of this substitution? Line 319 mentions that, compared to diffusion flow matching, DD “has an easy way to jump back to any point in the trajectory.” This is noted as a characteristic, but what practical benefits does it provide? Additionally, the ODE sampling trajectory in CM is also theoretically fixed and reversible. If my understanding is incorrect, please correct me.

Q2: Is there a way for this method to be trained from scratch, similar to Consistency Training?

Q3: Currently, the method requires a dataset size of 1.2M. How significantly does the data volume impact performance? Related distillation methods, such as DMD[1], use relatively smaller datasets. Is there a possibility of a similar approach to DMD2[2] that could discard the need for noise-data pairs?

Q4: Q: In line 234, is $\pi$ the same as z used later on? There may be a misalignment in the notation here.

[1]. Yin T, Gharbi M, Zhang R, et al. One-step diffusion with distribution matching distillation[C]//Proceedings of the IEEE/CVF Conference on Computer Vision and Pattern Recognition. 2024: 6613-6623.
[2]. Yin T, Gharbi M, Park T, et al. Improved Distribution Matching Distillation for Fast Image Synthesis[J]. arXiv preprint arXiv:2405.14867, 2024.

---

> ### Author Response · Authors · 2024-11-24
> **Response to Reviewer 2cqS 1**
>
> Thank you very much for your positive feedback! Here are our responses to your questions:
> - W1: results still fall short compared to the pre-trained model.
>   - We found out that by fixing the incorrect generation configurations and using more training iterations, the sample quality of DD can be substantially improved, getting much closer to the pre-trained model. (1) We first **fix the generation configuration of LlamaGen model** by simply modifying the suboptimal default sampling configuration of the pre-trained model (please see our response to reviewer ziid Q3 for details), which brings an improvement from FID=6.53 to FID=4.11, enabling a higher quality of the teacher distribution. (2) Then, we simply **train both VAR-DD and LlamaGen-DD for more epochs**. These modifications bring significant improvement to the quality of the few-step DD. Specifically, for LlamaGen, the FIDs of 1-step generation and 2-step generation improve from 17.98 and 11.24 to 11.35 and 7.58, and the FID of 81-step generation with pre-trained model also decreases from 10.30 to 5.71, which is very close to the pre-trained model; for VAR, the FID of 1 step generation improves from 10.65 to 9.94, and the FID of 6-step generation improves from 5.90 to 5.03. These results are updated in Table 1 in the revision. Additionally, we update the visualization examples in our revision (please see Fig.3, Fig.8-Fig.11 in our revision), which are of higher quality.
>   - Additionally, DD provides a flexible trade-off between time and quality by involving the teacher model. As shown in Table 2, we can replace the DD model with the pre-trained teacher model in some positions to get better generation quality. Our best results are already close to the pre-trained model (i.e., 4.20 v.s. 5.03 for VAR, 4.11 v.s. 5.71 for LlamaGen), while still having an acceleration ratio of 1.5$\times$ and 2.9$\times$.
>   - We believe that these updated results could address concerns about the sample quality of DD.
> - Q1.1: the advantage of replacing the diffusion model in CM with an AR model
>   - Auto-regressive (AR) models and diffusion models are two different families of generative models. While diffusion models have achieved remarkable success in visual generation, some studies have shown that AR models surpass diffusion models in this field [1][2]. Moreover, most mainstream large language models are AR models. These suggest that AR models might possess unique advantages over diffusion models. Therefore, we believe that the advantages of DD, as an accelerated AR model, compared to CM, as an accelerated diffusion model, could be partially attributed to the advantages of AR over diffusion models. For instance, theoretically speaking, DD has the potential to **handle discrete sequences like text**, and DD may also **achieve better generation quality**. As an exploratory first step in the few-step AR field, DD leaves significant room for future research.
>
>     [1] Tian et al., Visual Autoregressive Modeling: Scalable Image Generation via Next-Scale Prediction
>
>     [2] Sun et al., Autoregressive Model Beats Diffusion: Llama for Scalable Image Generation
>
> - Q1.2 & Q1.3: the benefits of having an easy way to jump back to any point in the trajectory & why CM doesn't have such property
>   - We want to point out that although the ODE trajectory is fixed and reversible, the reversion of the ODE trajectory requires **additional model inference**, because the score of the $x_t$ could not be obtained easily and can only be given by a diffusion model. An alternative method is to directly add noise to the target timestep (like the multi-step sampling method in CM). However, in this way, there is no guarantee that the landing point will still be on the \emph{current trajectory}. Such a problem does not exist in DD, since DD can jump back to any point in the \emph{current trajectory} by simply replacing the data at corresponding positions with the initial noise, without any model inference.
>   - Such property may be useful in some cases that require consistency when editing one sample. For example, if the user wants to polish a generated image without changing its overall content, DD can easily accomplish this without additional model inference to jump back along the current trajectory.
>
> - Q2: is there a way for this method to be trained from scratch, similar to consistency training?
>   - We also think that this is a very interesting question, and currently, we do not have a solution yet. We have listed it in Section 6 as an interesting and important future work.

---

> ### Author Response · Authors · 2024-11-24
> **Response to Reviewer 2cqS 2**
>
> - Q3: the possibility to discard the need for data-noise pairs
>   - Thank you for pointing out these interesting works to us! We first conduct ablation experiments to investigate the impact of dataset size on DD, with results of FID on 5k images shown below. We can see that with a smaller dataset size, DD can still work well.
> | | | | | | | | |
> |-|-|-|-|-|-|-|-|
> || | |1step generation | | | | |
> |Dataset Size|Epoch 10|Epoch 20|$\quad$$\quad$Epoch 30|Epoch 40|Epoch 50|Epoch 130| |
> |0.6M|27.37|23.32|$\quad$$\quad$20.85|19.73|19.10|-| |
> |0.9M|27.11|22.74|$\quad$$\quad$20.79|19.66|19.00|-| |
> |1.2M (default)|26.92|22.74|$\quad$$\quad$20.83|19.65|18.97|16.44| |
> |1.6M|25.64|22.25|$\quad$$\quad$20.63|19.32|18.75|-| |
> || | |2step generation | | | | |
> |0.6M|16.91|15.46|$\quad$$\quad$14.88|15.01|14.96|-| |
> |0.9M|16.20|15.52|$\quad$$\quad$14.97|14.74|14.63|-| |
> |1.2M (default)|16.43|15.45|$\quad$$\quad$14.96|14.81|14.57|14.12| |
> |1.6M|15.71|14.85|$\quad$$\quad$14.35|14.18|14.06|-| |
> | | | | | | | | |
>   - Regarding the relationship with DMD: DMD and DMD2 are novel diffusion distillation methods. Unlike other diffusion distillation methods [3][4][5][6] based on the consistency of ODE-trajectory, the key idea of DMD and DMD2 is distribution matching, which fundamentally eliminates the need for training data generation. We think such methods can also be applied to few-step AR distillation. From a high-level perspective, the noisy image in DMD is similar to the partial data sequence in AR, while the next token logits given by the pre-trained AR model can be viewed as the score function in DMD. The objective can be set as minimizing the distance between the output of a fake logits prediction network and the pre-trained AR at all timesteps. As long as the next-token conditional distribution of the generated distribution is the same as the distribution given by the pre-trained AR model, the modeled one-step distribution will be correct. There is no requirement for any data-noise pair in this case. Note that this idea is substantially different from our method, so we leave it as an interesting direction for future works.
>   - We added these discussions in our revision. Please check Appendix D. Thanks for pointing out this interesting direction!
>
>     [3] Luhman et al., Knowledge distillation in iterative generative models for improved sampling speed
>
>     [4] Salimans et al., Progressive distillation for fast sampling of diffusion models
>
>     [5] Song et al., Consistency models
>
>     [6] Kim et al., Consistency trajectory models: Learning probability flow ode trajectory of diffusion
> - Q4: notation misalignment in Line 234
>   - Thank you for pointing out our typo. The $\pi_0$ and $\pi_1$ here equal to the marginal probability distribution of $z_0$ and $z_1$ at $t=0$ and $t=1$, respectively. We have fixed this in our revision.

---

> > ### Comment · Reviewer_2cqS · 2024-11-25
> >
> > Thank you very much for the author’s response; my questions have been clearly answered. I have one last question, will the code be published?

---

> > > ### Author Response · Authors · 2024-11-25
> > >
> > > Thank you for your response! We plan to release both our code and checkpoints.

---

> > > > ### Comment · Reviewer_2cqS · 2024-11-25
> > > >
> > > > Really nice. I prefer to keep the score of 8 as a very good paper.

---

> > > > > ### Author Response · Authors · 2024-11-25
> > > > >
> > > > > Thank you for your positive feedback! We are opening to discussion if you have additional questions.

---

### Official Review · Reviewer_ziid · 2024-11-03

**Soundness:** 3
**Presentation:** 3
**Contribution:** 3
**Rating:** 6
**Confidence:** 4

**Summary:**

This paper introduces Distilled Decoding (DD), a novel method to reduce the sampling steps of autoregressive image generation models. The approach reframes the traditional next-token or scale prediction process into a denoising procedure guided by flow matching. Starting from fully noisy tokens, DD employs an autoregressive process to map these tokens into image tokens in a flexible number of sampling steps. Experiments with the LlamaGen and VAR models demonstrate that DD significantly outperforms conventional few-step sampling baselines.

**Strengths:**

- The authors proposed a new framework to make few-step AR distillation possible.
- The conversion from next image token prediction to next (set of) image token denoising is a very smart design. The method naturally combines the best of worlds in diffusion models / flow matching and autoregressive image modeling.
- The performance gain against baseline few-step samplers is huge.

**Weaknesses:**

- The authors propose a novel framework enabling few-step autoregressive (AR) distillation.

- Converting next-image-token prediction to next-image-token denoising is an ingenious design choice, seamlessly integrating the strengths of both diffusion models and autoregressive image modeling.

- The performance improvement over baseline few-step samplers is substantial.

**Questions:**

- The writing is somewhat unclear. It's not obvious whether $q_t$ represents tokens in the discrete codebooks or outcomes of flow matching. Algorithms 1-4 are much clearer than the main text, so refining the writing to enhance intuitiveness would benefit the readers.

- The performance gap between few-step AR samplers and the baseline remains quite large.

- Some baseline values, such as VAR/LlamaGen in Table 1, appear to be significantly lower than the original paper's reported results.

- It would be interesting to explore how DD performs in text-conditioned image generation tasks.

- The structure of DD closely resembles DMD in diffusion distillation, where a <noise, clean image> dataset is constructed; however, DMD isn't addressed in this paper. Adding a discussion on DMD would be beneficial. Additionally, transforming next-token prediction to AR-based token denoising at a high level seems conceptually similar to MAR (Li et al., 2024, cited). Including further discussion on MAR could also be helpful.

===

Update: I've read the author response and it has addressed my concerns, I will therefore keep my score and recommend acceptance.

---

> ### Author Response · Authors · 2024-11-24
> **Response to Reviewer ziid 1**
>
> Thank you very much for your positive feedback! Here are our responses to your questions:
> - Weakness: It seems that there might be typos in the weakness section: all weaknesses are actually the strengths of our work. If the reviewer has other concerns, please let us know and we are willing to discuss them!
> - Q1: unclear writing. For example, unclear representation of $q_t$
>   - $q_t$ refers to both *tokens in the discrete codebooks* and *outcomes of flow matching*. For flow-matching, after completing the ODE solving process, we look up the token closest to the output from all tokens in the codebook, to enable a strict alignment between flow-matching-based sampling and discrete sampling. As a result, the flow-matching output is also a token in the codebook. We clarify it in the revision. Thank you for pointing it out!
> - Q2: performance gap between few-step AR samplers and the baseline
>   - We found out that by fixing the incorrect generation configurations and using more training iterations, the sample quality of DD can be substantially improved, getting much closer to the pre-trained model. (1) We first **fix the generation configuration of LlamaGen model** by simply modifying the suboptimal default sampling configuration of the pre-trained model (please see our response to your Q3 for details), which brings an improvement from FID=6.53 to FID=4.11, enabling a higher quality of the teacher distribution. (2) Then, we simply **train both VAR-DD and LlamaGen-DD for more epochs**. These modifications bring significant improvement to the quality of the few-step DD. Specifically, for LlamaGen, the FIDs of 1-step generation and 2-step generation improve from 17.98 and 11.24 to 11.35 and 7.58, and the FID of 81-step generation with pre-trained model also decreases from 10.30 to 5.71, which is very close to the pre-trained model; for VAR, the FID of 1 step generation improves from 10.65 to 9.94, and the FID of 6-step generation improves from 5.90 to 5.03. These results are updated in Table 1 in the revision. Additionally, we update the visualization examples in our revision (please see Fig.3, Fig.8-Fig.11 in our revision), which are of higher quality.
>   - Additionally, DD provides a flexible trade-off between time and quality by involving the teacher model. As shown in Table 2, we can replace the DD model with the pre-trained teacher model in some positions to get better generation quality. Our best results are already close to the pre-trained model (i.e., 4.20 v.s. 5.03 for VAR, 4.11 v.s. 5.71 for LlamaGen), while still having an acceleration ratio of 1.5$\times$ and 2.9$\times$.
>   - We believe that these updated results could address concerns about the sample quality of DD.
> - Q3: baseline values lower than the original paper
>   - Thanks for catching these discrepancies! We checked these results again.
>   - For LlamaGen, initially, we followed the official generation script (https://github.com/FoundationVision/LlamaGen/blob/main/autoregressive/sample/sample_c2i.py) to generate the samples and got the FID 6.53 we reported in the submission. Later, we realized that the other official generation script with DDP support (https://github.com/FoundationVision/LlamaGen/blob/main/autoregressive/sample/sample_c2i_ddp.py) used a different top_k setting. We followed this new script by setting a larger top_k and got an FID of 4.11, which is within the reasonable range compared to the officially reported result 3.80. We have updated all our LLamaGen experiments using this new generation configuration.
>   - For VAR, we just follow the default sampling setting in https://github.com/FoundationVision/VAR/blob/main/demo_sample.ipynb (i.e., topk=900, top_p=0.95), and apply CFG=2.0 following the paper. However, we were only able to reproduce an FID score of 4.19. We think this may be because of the different FID calculation codebase: we use PytorchFID (see https://github.com/mseitzer/pytorch-fid) to calculate FID while VAR did not release their evaluation code. As discussed in [1], different FID calculation codebases may introduce some gaps in the number. We want to further point out two facts:
>     - **The unalignment is not caused by the flow-matching sampler.** We run the original multinomial sampler and get similar results, as shown below.
> | | | | | | | | |
> |-|-|-|-|-|-|-|-|
> |Sampler|Flow-matching|Multinomial seed1|Multinomial seed2| | | | |
> |FID|4.19|4.16|4.26| | | | |
> | | | | | | | | |
>     - In our paper, **we use the same FID codebase for the evaluation of all methods, so the comparison is fair.**
>
>    [1] Parmar et al., On Aliased Resizing and Surprising Subtleties in GAN Evaluation

---

> ### Author Response · Authors · 2024-11-24
> **Response to Reviewer ziid 2**
>
> - Q4: It would be interesting to explore how DD performs in T2I tasks
>   - Thank you for the great suggestions! Following your suggestion, we conduct the following experiments.
>   - **Experiment settings.** We use the official T2I-XL-Stage1 model of LlamaGen (https://huggingface.co/peizesun/llamagen_t2i/resolve/main/t2i_XL_stage1_256.pt) (pre-trained on LAION-COCO) as the pre-trained AR model to be distilled. We generate 1.9M tuples of (text, noise, and data) to train our DD model.
>   - **Results.** Note that T2I is a very difficult task that requires a large amount of data and training costs. Due to the time constraints of the discussion period and many other experiments we added in the rebuttal, we only managed to conduct a small amount of training (around 15 epochs). However, even with such limited training, the DD model already achieves reasonable results. We report the FID scores (evaluated using 5000 generated images) below. As a comparison, the FID of the pre-trained model with 256-step sampling is 25.70. We can see that: (1) the performance of 2-step DD is already close to the pre-trained model (29.90 vs. 25.70), while reducing the number of generation steps by 128 times; (2) The performance of DD is still improving stably at the end of 15 epochs, so we expect that the final performance could be even better. The generated images can be found at https://anonymous.4open.science/r/DD_t2i_ano-F222 .
> | | | | | | | | |
> |-|-|-|-|-|-|-|-|
> | |100k|200k|300k|400k|460k (~15 epochs)| | |
> |1 step|51.66|43.21|41.17|40.05|39.04| | |
> |2 step|56.31|34.12|31.75|30.66|29.90| | |
> | | | | | | | | |
>   - Although the T2I training is not yet complete (and likely will not be finished before the discussion period ends), we believe our current methods and results already represent a solid contribution. **(1)** 1-step generation of image AR models is a challenging problem, even for non-T2I (i.e., label-to-image generation) tasks. We are not aware of, even among concurrent ICLR submissions, any approaches that can do 1-step generation of image AR models. The reason is fundamental--as we discussed in Section 3.1, existing methods based on vanilla AR are fundamentally impossible to achieve good 1-step generation. We believe that DD, as the first method that enables reasonable 1-step generation of AR models, makes non-trivial and solid contributions to the field.  **(2)** On top of that, we already got promising and non-trivial T2I results even with limited training iterations. We believe that the current results have already demonstrated the potential of DD to be applied in such tasks.
>   - We will update the T2I results in the revision once the training is converged.
> - Q5: Lacking discussions on related works, like DMD and MAR.
>   - Thank you for pointing out these related works to us!
>   - DMD uses distribution matching to distill diffusion models into few steps. Specifically, the main idea of DMD is to minimize the distance between the generated distribution and the teacher distribution at all timesteps. Additionally, it constructs data-noise pairs and conducts direct distillation as a regularization loss term to avoid mode collapse. There are several key differences between DD and DMD: **(1) Target:** DMD focuses on diffusion models, whereas DD focuses on AR. These are two different problems; **(2) Technique:** In diffusion models, the diffusion process naturally defines the data-noise pairs, which can be directly used for distillation as in DMD. In contrast, AR models do not have a pre-defined concept of *noise*. How to construct noise and the data-noise pair (Section 3.2) is one important and unique contribution of our work. That being said, the distribution matching idea in DMD is very insightful, and could potentially be combined with DD to achieve better approaches for few-step sampling of AR models.
>   - MAR proposes to replace the cross-entropy loss with diffusion loss in AR, which shares some similarities with DD at a high level since both works view decoding as a denoising process. The goals of these two works are different though. MAR targets removing the vector quantization in image AR models for better performance, while DD aims to compress the sampling steps of AR, without any modification to the codebook. In this sense, these two works are orthogonal and could potentially be combined to develop a new method that leverages the strengths of both.
>   - We have added these discussions to the revision.

---

> ### Author Response · Authors · 2024-11-28
> **Follow up Comment**
>
> Dear Reviewer ziid,
>
> We would like to know if you have additional questions. We are happy to discuss them further. If our response has addressed your concerns, would you mind updating the score to reflect that?
>
> Thank you once again for your time and the insightful questions, which were very helpful in improving the quality of this paper.

---

> ### Author Response · Authors · 2024-12-03
> **Follow-up Comment**
>
> Dear reviewer ziid,
>
> As the deadline of the discussion phase is approaching, we kindly ask you to review our response and see whether it can address your concerns. We have followed your suggestion and updated the sampling method of the baseline approach. Then we updated the main experimental results of DD based on this. Additionally, we have included experiments on text-to-image task. If our response has addressed your concerns, would you mind reconsidering your score?
>
> Thank you once again for your effort and valuable feedback.

---

### Official Review · Reviewer_cVWg · 2024-11-04

**Soundness:** 3
**Presentation:** 2
**Contribution:** 3
**Rating:** 6
**Confidence:** 4

**Summary:**

The method maps autoregressive model output distributions to a Gaussian distribution through flow matching, creating a deterministic transformation. The goal is to then learn this mapping with a neural network. This enables parallel token generation while preserving the conditional dependencies between tokens in the AR model's output distribution.

**Strengths:**

The work presents a method that leverages deterministic flow matching to create training data (from an AR model) for a one-step image generation model. When trained on this data this model is a distilled version of the original AR model. The idea of using determinstic flow matching to create the data is novel and seems like a good and innovative candidate idea to achieve this. The paper evaluates the claims on class-to-image generation on ImageNet and compares to simple baselines, achieving acceptable FID scores, that are high but not too high.

**Weaknesses:**

A. It seems like the FID increases, although seemingly acceptable in numerical terms, give rise to blurry and artifact-ridden images, many of which don't even preserve the structure of the class they are trying to generate (monkeys without eyes etc.). Also, another thing that undermines my confidence is that no images are shown in the main paper and instead shown in the appendix. At least some examples are shown.

B. One big problem from (A) is that, since the paper's main premise is to distill an AR model into a one-shot model, good samples are required to demonstrate that this approach is viable. Given the current results we do not know if this approach can actually generate satisfactory images even when scaled or taken into another regime with larger data such as T2I.

I think these are the main issues for me to be convinced of this approach. My question to authors are:
- Do you expect the approach to become good enough to generate satisfactory images with different scale (data, model size)?
- If so, what leads you to believe this?

**Questions:**

1. Your method depends on DPM-Solver to generate the trajectories. Do you think this affects results in any way? What other dependencies does your method have that might affect results?
2. Do the hyperparameters for DPM-Solver affect the results drastically? Why did you pick the current hyperparams?
3. Do you think a larger amount of timesteps might be needed to get satisfactory results? Is this compatible with your method?
4. Skip baselines seem conceptually weak. The one-step baseline is guaranteed to fail given results on toy examples. Are there no better baselines - for example progressive distillation, that is used for diffusion models?

---

> ### Author Response · Authors · 2024-11-24
> **Response to Reviewer cVWg 1**
>
> Thank you very much for your valuable suggestions! We added several experiments according to your suggestions. Here are the details.
>
> - W1.1: FID and sample quality are not good enough.
>   - We found out that by fixing the incorrect generation configurations and using more training iterations, the sample quality of DD can be substantially improved, getting much closer to the pre-trained model. (1) We first **fix the generation configuration of LlamaGen model** by simply modifying the suboptimal default sampling configuration of the pre-trained model (please see our response to reviewer ziid Q3 for details), which brings an improvement from FID=6.53 to FID=4.11, enabling a higher quality of the teacher distribution. (2) Then, we simply **train both VAR-DD and LlamaGen-DD for more epochs**. These modifications bring significant improvement to the quality of the few-step DD. Specifically, for LlamaGen, the FIDs of 1-step generation and 2-step generation improve from 17.98 and 11.24 to 11.35 and 7.58, and the FID of 81-step generation with pre-trained model also decreases from 10.30 to 5.71, which is very close to the pre-trained model; for VAR, the FID of 1 step generation improves from 10.65 to 9.94, and the FID of 6-step generation improves from 5.90 to 5.03. These results are updated in Table 1 in the revision. Additionally, we update the visualization examples in our revision (please see Fig.3, Fig.8-Fig.11 in our revision), which are of higher quality.
>   - Additionally, DD provides a flexible trade-off between time and quality by involving the teacher model. As shown in Table 2, we can replace the DD model with the pre-trained teacher model in some positions to get better generation quality. Our best results are already close to the pre-trained model (i.e., 4.20 v.s. 5.03 for VAR, 4.11 v.s. 5.71 for LlamaGen), while still having an acceleration ratio of 1.5$\times$ and 2.9$\times$.
>   - We believe that these updated results could address concerns about the sample quality of DD.
> - W1.2: Show images in the main paper.
>   - Due to space constraints, we were not able to fit the generated images in the main paper and deferred them to the appendix. We understand the importance of showing visualization in the main paper. Following your suggestions, we move some visualizations to the main paper in Fig.3.
> - W2.1: Results on different model sizes.
>   - In the initial submission, we reported results on two models: one for LlamaGen and one for VAR. Following the suggestions, **we have added experiments on three additional models, bringing the total to five**: two model sizes of LlamaGen and three model sizes of VAR. The results are shown in the following table. The key takeaways are:
>     - **DD achieves reasonable sample quality across different model sizes**: Across all five models, FID is in the range of 8.92-11.35 for one-step generation and 6.95-11.17 for two-step generation.
>     - **DD scales well with model sizes**: for each model family, with larger model sizes, DD can achieve better FID.
>   - These results suggest that DD works and scales well with different model sizes.
> | | | | | | | | |
> |-|-|-|-|-|-|-|-|
> |Type|Method|#Param|#Step|FID|IS|Prec|Recall|
> || | $\quad$$\qquad$VAR | | | | | |
> |Baseline|VAR-d16|310M|10|4.19|230.2|0.84|0.48|
> |Baseline|VAR-d20|600M|10|3.35|301.4|0.84|0.51|
> |Baseline|VAR-d24|1.03B|10|2.51|312.2|0.82|0.53|
> |Ours|VAR-d16-DD|327M|1|9.94|193.6|0.80|0.37|
> |Ours|VAR-d16-DD|327M|2|7.82|197.0|0.80|0.41|
> |Ours|VAR-d20-DD|635M|1|9.55|197.2|0.78|0.38|
> |Ours|VAR-d20-DD|635M|2|7.33|204.5|0.82|0.40|
> |Ours|VAR-d24-DD|1.09B|1|8.92|202.8|0.78|0.39|
> |Ours|VAR-d24-DD|1.09B|2|6.95|222.5|0.83|0.43|
> || |$\qquad$LlamaGen | | | | | |
> |Baseline|LlamaGen-B|111M|256|5.42|193.5|0.83|0.44|
> |Baseline|LlamaGen-L|343M|256|4.11|283.5|0.85|0.48|
> |Ours|LlamaGen-B-DD|98.3M|1|15.50|135.4|0.76|0.26|
> |Ours|LlamaGen-B-DD|98.3M|2|11.17|154.8|0.80|0.31|
> |Ours|LlamaGen-L-DD|326M|1|11.35|193.6|0.81|0.30|
> |Ours|LlamaGen-L-DD|326M|2|7.58|237.5|0.84|0.37|

---

> ### Author Response · Authors · 2024-11-24
> **Response to Reviewer cVWg 2**
>
> - W2.2: Results on text-to-image generation
>   - Thank you for the great suggestions! Following your suggestion, we conduct the following experiments.
>   - **Experiment settings.** We use the official T2I-XL-Stage1 model of LlamaGen (https://huggingface.co/peizesun/llamagen_t2i/resolve/main/t2i_XL_stage1_256.pt) (pre-trained on LAION-COCO) as the pre-trained AR model to be distilled. We generate 1.9M tuples of (text, noise, and data) to train our DD model.
>   - **Results.** Note that T2I is a very difficult task that requires a large amount of data and training costs. Due to the time constraints of the discussion period and many other experiments we added in the rebuttal, we only managed to conduct a small amount of training (around 15 epochs). However, even with such limited training, the DD model already achieves reasonable results. We report the FID scores (evaluated using 5000 generated images) below. As a comparison, the FID of the pre-trained model with 256-step sampling is 25.70. We can see that: (1) the performance of 2-step DD is already close to the pre-trained model (29.90 vs. 25.70), while reducing the number of generation steps by 128 times; (2) The performance of DD is still improving stably at the end of 15 epochs, so we expect that the final performance could be even better. The generated images can be found at https://anonymous.4open.science/r/DD_t2i_ano-F222 .
> | | | | | | | | |
> |-|-|-|-|-|-|-|-|
> | |100k|200k|300k|400k|460k (~15 epochs)| | |
> |1 step|51.66|43.21|41.17|40.05|39.04| | |
> |2 step|56.31|34.12|31.75|30.66|29.90| | |
>   - Although the T2I training is not yet complete (and likely will not be finished before the discussion period ends), we believe our current methods and results already represent a solid contribution. **(1)** 1-step generation of image AR models is a challenging problem, even for non-T2I (i.e., label-to-image generation) tasks. We are not aware of, even among concurrent ICLR submissions, any approaches that can do 1-step generation of image AR models. The reason is fundamental--as we discussed in Section 3.1, existing methods based on vanilla AR are fundamentally impossible to achieve good 1-step generation. We believe that DD, as the first method that enables reasonable 1-step generation of AR models, makes non-trivial and solid contributions to the field.  **(2)** On top of that, we already got promising and non-trivial T2I results even with limited training iterations. We believe that the current results have already demonstrated the potential of DD to be applied in such tasks.
>   - We will update the T2I results in the revision once the training is converged.
> - W2.3: Results on different data scales.
>   - Following your suggestions, we vary the number of the (data, noise) pairs used to train DD to evaluate its scaling ability towards more data.
>   - **Experiment settings.**  We test the performance of DD under different dataset sizes, including 0.6M, 0.9M, 1.2M, and 1.6M. Due to the limitation of time and resources, we only train every model for 50 epochs and report the 1-step and 2-step FID with 5k generated images. The results are shown in the following table. **The key takeaway is that the performance of DD improves consistently with more training data, indicating its good scalability with respect to data.**
> | | | | | | | | |
> |-|-|-|-|-|-|-|-|
> || | |1step generation | | | | |
> |Dataset Size|Epoch 10|Epoch 20|$\quad$$\quad$Epoch 30|Epoch 40|Epoch 50|Epoch 130| |
> |0.6M|27.37|23.32|$\quad$$\quad$20.85|19.73|19.10|-| |
> |0.9M|27.11|22.74|$\quad$$\quad$20.79|19.66|19.00|-| |
> |1.2M (default)|26.92|22.74|$\quad$$\quad$20.83|19.65|18.97|16.44| |
> |1.6M|25.64|22.25|$\quad$$\quad$20.63|19.32|18.75|-| |
> || | |2step generation | | | | |
> |0.6M|16.91|15.46|$\quad$$\quad$14.88|15.01|14.96|-| |
> |0.9M|16.20|15.52|$\quad$$\quad$14.97|14.74|14.63|-| |
> |1.2M (default)|16.43|15.45|$\quad$$\quad$14.96|14.81|14.57|14.12| |
> |1.6M|15.71|14.85|$\quad$$\quad$14.35|14.18|14.06|-| |

---

> ### Author Response · Authors · 2024-11-24
> **Response to Reviewer cVWg 3**
>
> - Q1 & Q2: The choice of DPM-Solver and how its hyperparameters affect the results.
>   - First of all, we want to clarify that, the DPM-Solver is only used to solve the ODE mapping between the noise tokens and the output tokens; it is *not* directly used in the final generation process.
>   - **The criterion of the ODE solver selection.** The ODE mapping we constructed in Section 3.2 theoretically ensures that the distribution of output tokens from the ODE process follows the ground-truth token distribution of the pre-trained teacher model. However, in practice, we cannot compute the ODE process precisely and need a numerical solver to approximate it. **Any ODE solver that has a small enough approximation error is applicable.**
>   - **Why we chose DPM-Solver.** DPM-Solver is the first and the only ODE solver we tried. We chose it as it is considered one of the state-of-the-art ODE solvers that requires a small number of sampling steps (so it is fast for us) and is used in popular models such as Stable Diffusion. We found that its approximation error is already small enough for our use case: using only 10 sampling steps, the FID of the samples generated from the ODE process is 4.20, close enough to  4.21$\pm$0.05, the FID of the ground-truth generation process using multinomial sampling.
>   - **The hyper-parameter choices.** The hyper-parameters of the DPM-Solver influence the approximation error, which thereby impacts the upper bound performance of DD. For most hyper-parameters, including order (3), timestep skip type (time_uniform), solver method (multistep), prediction type (data prediction), and lower final order (True), we use the default values of DPM-Solver. The only exceptions are the start epsilon and the end epsilon, which define the actual starting and ending timestep of the solving process by a smal offsets to ensure numerical stability. We need to set different values for these two parameters, as the noise schedule in Rectified Flow (which we used to construct the mapping between noise and output tokens) is different from the one used in DPM-Solver (i.e., VP and VE SDE). We tried several values (see the table below) and picked the one (i.e., 1e-4/5e-2) that has minimal degradation on the FID score compared to the original multinomial sampling (i.e., lowest approximation error).
> | | | | | | | | |
> |-|-|-|-|-|-|-|-|
> |Eps start/end|1e-4/1e-4|1e-4/1e-3|1e-4/1e-2|1e-4/5e-2|1e-3/5e-2|Multinomial| |
> |FID-5k|145.5|123.9|10.38|10.27|10.33|10.26| |
>   - Please let us know if anything is still unclear to you, and we will be happy to clarify further.
> - Q3: How the number of timesteps affect the results
>   - We are sorry that we do not fully understand what you mean by "timesteps" here. Do you mean the number of timesteps used **in the DPM-Solver**, or **in the DD sampling process**?
>   - If you mean **in the DPM-Solver**: Please refer to our answers to Q1&Q2. We used 10 timesteps and found that it already had minimal degradation on the FID score, so we did not try other values.
>   - If you mean **in the DD sampling process**: Beyond 1-step and 2-step generation quality in Table 1, we further presented the trade-off between more sampling steps and the generation quality (e.g., FID scores) in Table 2. Our best results are already close to the pre-trained model (i.e., 4.20 v.s. 5.03 for VAR, 4.11 v.s. 5.71 for LlamaGen), while still having an acceleration ratio of 1.5$\times$ and 2.9$\times$.

---

> ### Author Response · Authors · 2024-11-24
> **Response to Reviewer cVWg 4**
>
> - Q4: The one-step baseline is guaranteed to fail given results on toy examples. Skip baselines are too weak.  Are there any other baselines such as progressive distillation that we can consider?
>   - Thank you for the question! We carefully considered the baseline during this work. *Given that we are **the first** to consider the challenging problem of 1-step generation for image AR models, there is no existing approach against which we can directly compare.* Therefore, we had to resort to creating baselines on our own. Below, we explain our baselines and the one the reviewer proposed.
>   - **The one-step baseline.** We want to clarify that this approach is not a "toy" case. As explained in Section 1, the most popular approach for reducing the generation steps of image AR models is to draw multiple tokens at a time. As derived in Section 3.1 and Appendix A, this one-step baseline represents the **optimal** solution for such methods under one-step generation, which means that the performance of a neural network is theoretically unable to surpass it. The fact that this baseline fails in one-step generation supports our key insights from Section 1 and Section 3.1: existing approaches are fundamentally incapable of one-step generation. Furthermore, it underscores the inherent challenges of achieving one-step generation. We consider these an important contribution of our work.
>   - **The skip baselines.** This is the most natural method we could think of that can achieve few-step generation directly from a pre-trained model. That being said, we agree with the reviewer that this baseline is conceptually weak, but this was the best approach we could think of.
>   - **Progressive distillation.** We are sorry that we are not fully sure if the reviewer means (1) Using progressive distillation to train few-step AR models, or (2) comparing progressive distillation applied to diffusion models.
>     - **(1) Using progressive distillation to train AR models.** Progressive distillation cannot be directly applied to AR models---if we naively do that, the last stage of progressive distillation would require the model to directly predict all tokens, which would run into the same problem discussed in Section 3.1 and the generation results are guaranteed to be bad. In order to apply progressive distillation to AR models, we would still need to construct a deterministic mapping from noise tokens to data tokens as in Section 3.2, which is one of the key contributions of DD. The resulting method should be considered as a follow-up work of DD instead of a baseline.
>     - **(2) Comparing progressive distillation applied on diffusion models.** AR and diffusion models are two different families of generative models. Since our work focuses on speeding up image AR models, we do not think comparing methods for speeding up diffusion models is necessary.
>   - If the reviewer has other suggestions on the baselines, we are happy to try them out!

---

> ### Comment · Reviewer_cVWg · 2024-11-25
>
> Q1. Yes I understand that it's not used in the final generation process. Thank you for all the other details, this clarifies the work for me.
>
> Q3. I mean in the DD sampling process. I would say results are still a little underwhelming qualitatively for 1 and 2-step but it's good to have the FIDs for other sampling steps.
>
> Q4: Thank you, this is a good answer to my concerns.

---

> > ### Comment · Reviewer_cVWg · 2024-11-25
> > **Increasing my score to 6**
> >
> > I think the authors did a great job in engaging with all of my concerns and convincing me that this work is (1) thorough and well-considered (2) novel and interesting for the community (3) achieves good results even though they are still a little underwhelming qualitatively they would improve given other advances or scaling.
> >
> > Thank you for the detailed work on the paper and the responses.

---

> > > ### Author Response · Authors · 2024-11-25
> > > **Thank you**
> > >
> > > Thank you once again for your insightful suggestions, which have been incredibly helpful in enhancing the quality of the paper!

---

### Author Response · Authors · 2024-11-24
**General Response to all Reviewers**

We sincerely thank all reviewers for their insightful comments and suggestions. We are delighted to see that the reviewers think **our method is novel (cVWg, ziid), smart (ziid), and interesting (2cqS, ni7B), the performance against few-step sampling baselines is huge (ziid, ni7B), and the paper is well-written (2cqS, ni7B).**

Below, we summarize the key questions from the reviews, each of which we have addressed with extensive experiments.

1. **The FID and sample quality are not good enough (cVWg, ziid, 2cqS, ni7B).** We found that all our reported results can be substantially improved and get much closer to the pre-trained model. For example, by simply fixing the generation configuration of the LlamaGen model (pointed out by Reviewer ziid) and training for longer epochs, **the FID of our LlamaGen-DD is improved from 17.98 to 11.35 for 1-step generation, 11.24 to 7.68 for 2-step generation, and 10.30 to 5.71 for 81-step generation. These improved results are much closer to the 256-step LlamaGen model with FID 4.11.** Please see the rebuttal and the revision for detailed results in other settings and models.

2. **Results on different model sizes (cVWg).** In the initial submission, we reported results on two models: one for LlamaGen and one for VAR. Following the reviewer's suggestions, **we have added experiments on three additional models, bringing the total to five: two model sizes of LlamaGen and three model sizes of VAR.** The results are consistent: **(1) DD achieves reasonable sample quality across different model sizes**: Across all five models, FID is in the range of 8.92-11.35 for one-step generation and 6.95-11.17 for two-step generation. **(2) DD scales well with model sizes**: with larger model sizes, DD can achieve better FID. Please see the rebuttal and the revision for detailed results.

3. **Results on different data scales (cVWg, 2cqS).** Following the reviewers' suggestions, we conduct experiments on training DD on different numbers of (data, noise) pairs. Please see the rebuttal and the revision for detailed results. The key takeaways are:
    1. We find that DD can still work well with a smaller dataset size. For example, with only half of the data, the FID only increase less than 0.5 at epoch 50 (i.e., from 18.97 to 19.10 for 1step generation, from 14.57 to 14.96 for 2step generation).
    2. Additionally, with more (data, noise) pairs, the performance of DD can improve. For instance, the 2step FID of 1.6M dataset size has already surpassed the final converged performance of 1.2M dataset size (14.06 v.s.14.12) at an early epoch. The results demonstrate DD's robustness to limited data and its scaling ability.

4. **Results on text-to-image version of LlamaGen (cVWg, ni7B, ziid).** Following the reviewers' suggestions, we conducted the experiments. On LAION-COCO dataset, DD achieves an FID score of **29.90** with **2-step** generation (calculated with 5k generated images), while the FID of the pre-trained model with **256-step** sampling is **25.70**. In addition, the generated images show that DD is able to follow the text instructions similar to the teacher model, with several examples shown in this anonymous link: https://anonymous.4open.science/r/DD_t2i_ano-F222 . Note that due to the constraints on the time and computation resources, we have only been able to finish around 15 epochs for now. According to the FID training curve shown in the table below, the training is still in the early stage and FID scores are still decreasing stably, so we expect that the final results will be much better. We will update the revision once the full training is done.

| | | | | | | | |
|-|-|-|-|-|-|-|-|
| |100k|200k|300k|400k|460k (~15 epochs)| | |
|1 step|51.66|43.21|41.17|40.05|39.04| | |
|2 step|56.31|34.12|31.75|30.66|29.90| | |

**In summary, following the reviewers' suggestions, we have conducted more than 39 new experiments, including 9 training experiments, 5 dataset generation processes, and more than 25 evaluation experiments.** Due to the large amount of experiments, one experiment (text-to-image generation) is not fully done. However, we believe that our current experiments can already address the main concerns of the reviewers. These results are updated in the revision in red.
If the reviewers have any further questions or suggestions, please feel free to let us know and we look forward to further discussions!

---

### Meta-Review · Area_Chair_dfHu · 2024-12-24

**Metareview:**

This paper presents a novel approach to distilling multi-step autoregressive (AR) image generation models into flow models capable of generating images in just one or a few steps. This method opens up new possibilities for distilling AR models and achieves a reasonable FID while substantially reducing inference computations. The key strengths of the work lie in its ability to one-step image generation for AR models as well as the independence of training data, a significant step forward for efficient generative modeling.

The primary weakness of the paper is the noticeable performance gap compared to the original AR models. However, considering the novelty of the method and the potential it holds for advancing the field, I believe this gap does not undermine its contribution. Considering the consistent positive feedback from reviewers and the promising direction this work sets for future research, I recommend accepting this paper.

**Additional Comments On Reviewer Discussion:**

The primary concern raised during the review process focused on the performance gap between the original AR model and the distilled flow model. While the provided 2-step generation results show improvement, they do not entirely bridge this gap. However, this limitation is outweighed by the contribution of this work as the first attempt to distill autoregressive models into efficient few-step generation.

---

### Decision · Program_Chairs · 2025-01-22

Accept (Poster)